# AdaMeZO: Adam-style Zeroth-Order Optimizer for LLM Fine-tuning Without Maintaining the Moments

**Zhijie Cai** [* 1 2 3]   **Haolong Chen** [* 1 2 3]   **Guangxu Zhu** [1 2 3 4]

## Abstract

Fine-tuning LLMs is necessary for various dedicated downstream tasks, but classic backpropagation-based fine-tuning methods require substantial GPU memory. To this end, a recent work, MeZO, which relies solely on forward passes to fine-tune LLMs, significantly reduces GPU requirements at the cost of slower convergence due to its indifference to loss landscapes. Standard solutions, such as Adam, explore loss landscapes by estimating the first- and second-order moments and storing them in memory to guide the model's movement through dimensions with lower curvature and vice versa. However, directly applying Adam negates MeZO's advantage as it will triple the memory requirement. In light of this, we propose AdaMeZO, a zeroth-order optimizer that leverages Adam-style first- and second-moment estimates without maintaining them in memory. We present a theoretical analysis of AdaMeZO, corroborated by extensive experiments demonstrating AdaMeZO's performance, showing that AdaMeZO can outperform MeZO while requiring up to $70\%$ fewer forward passes. Trajectory visualizations affirm AdaMeZO's ability to adapt to diverse loss landscapes.

## 1. Introduction

Fine-tuning LLMs is necessary for specialized downstream tasks and has recently attracted significant attention. Many works have emerged that aim to tune models while accessing as little memory as possible. Popular first-order meth-

ods known as parameter-efficient fine-tuning (PEFT) to alleviate the heavy memory cost by modifying only a small (potentially extra) part of the whole model (Hu et al., 2022; Li & Liang, 2021; Lester et al., 2021; Dettmers et al., 2023; Pan et al., 2024). Additionally, a zeroth-order method (Malladi et al., 2023) enables discarding backpropagation, the primary contributor to LLM fine-tuning's memory cost, making it accessible on resource-limited devices.

As shown in Table 1, MeZO features an SGD-style update rule (Rumelhart et al., 1986; Bottou et al., 2018), allowing in-place parameter modification. After in-place model perturbation for gradient projection estimation, the gradients are not dumped into memory; instead, they are generated by a pseudo-random number generator (PRNG) before being scaled by the previously computed projection, reducing the memory cost for fine-tuning to the equivalent of deploying one. However, updating the model with only the most recent gradient estimate can lead to slower convergence, especially with noisy, isotropic zeroth-order gradient estimators. In comparison, adaptive optimizers like Adam (Kingma & Ba, 2014) and AdamW (Loshchilov & Hutter, 2017), which correct updates with preconditioners, are more widely adopted since the loss landscapes of LLMs exhibit complex curvature spectra across different dimensions, as documented in (Sagun et al., 2016; Ghorbani et al., 2019; Zhang et al., 2023; Das et al., 2024).

However, adaptive optimizers retain historical gradient information in the memory. In the case of Adam, the first and second moments are the accumulated gradients and the quadratic gradients. In other words, two vectors of the same size as the model need to be kept in memory. Considering that first-order methods use backpropagation, the additional memory cost is relatively small. But in the context of zeroth-order optimizers, the memory cost is multiplied.

Adaptive zeroth-order optimizers for LLM fine-tuning have recently attracted research interest, as shown in Table 1. Pioneering works include HiZOO (Zhao et al., 2024b), ZO-AdaMU (Jiang et al., 2024), and Helene (Zhao et al., 2024a). HiZOO proposes approximating the diagonal Hessian using an additional forward-pass oracle, which doubles the memory required to store it. Helene is a more direct integration of zeroth-order gradient estimation and

*Equal contribution   [1]Shenzhen International Center for Industrial and Applied Mathematics [2]Shenzhen Research Institute of Big Data [3]The Chinese University of Hong Kong-Shenzhen [4]Shenzhen Loop Area Institute. Correspondence to: Guangxu Zhu <gxzhu@sribd.cn>.

*Proceedings of the 43rd International Conference on Machine Learning*, Seoul, South Korea. PMLR 306, 2026. Copyright 2026 by the author(s).

*Table 1.* Key features for AdaMeZO and methods in comparison. $P$ in the first column denotes the amount of memory required to store the model weight, and $B \gg P$ denotes the amount of memory required to perform backpropagation. $\delta \ll 1$ is a small positive number. "FP" abbreviates forward pass.

|  | Param. memory | FP per step | 1st moment | 2nd moment |
|---|---|---|---|---|
| Adam (Kingma & Ba, 2014) | $3P + B$ | 1 | ✓ | ✓ |
| MeZO (Malladi et al., 2023) | $P$ | 2 | ✗ | ✗ |
| HELENE (Zhao et al., 2024a) | $3P$ | 2 | ✓ | ✓ |
| HiZOO (Zhao et al., 2024b) | $2P$ | 3 | ✗ | ✓ |
| AdaMeZO | $(\mathbf{1} + \delta)\mathbf{P}$ | **2** | ✓ | ✓ |

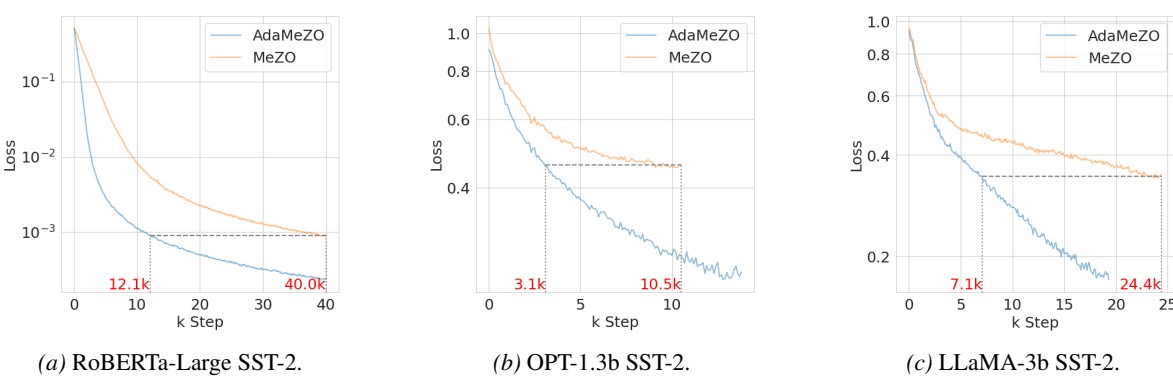

*(a)* RoBERTa-Large SST-2.      *(b)* OPT-1.3b SST-2.      *(c)* LLaMA-3b SST-2.

*Figure 1.* Loss curves of MeZO and AdaMeZO on the SST2 task. When fine-tuning RoBERTa-large, OPT-1.3b, LLaMA-3b, AdaMeZO took $69.75\%, 70.48\%, 70.90\%$ fewer forward passes to reach the loss values of MeZO at terminations, respectively. Hyperparameters and terminal conditions are detailed in Section B.4.

an Adam optimizer, and ZO-AdaMU replaces the moments with an uncertain version. As a result, the memory requirement is tripled to store both diagonal Hessian estimation and cumulative history gradients. However, despite the substantial increase in memory cost, they still use much less memory than first-order approaches and achieve a noticeable performance gain over MeZO.

In light of the above, we introduce AdaMeZO[1], a zeroth-order optimizer that leverages Adam-style first- and second-moment estimates to accelerate convergence without requiring additional memory to store them. This is made possible by 1) computing truncated moments that discard outdated gradients rather than faithfully maintaining the full moment estimations, and 2) block-wise generation of random gradient direction with a finer operation of the PRNG. As a result, AdaMeZO significantly reduces the number of forward passes required for convergence and improves the fine-tuned model's performance. A summary of the contributions of this work is as follows.

1. We introduce AdaMeZO, an optimizer that uses zeroth-order gradient estimates and updates with Adam-style first- and second-moment estimates. Although the moments are necessary to compute the

model updates, with truncated approximations and finer PRNG operations, they do not need to be stored in memory. In this way, AdaMeZO can theoretically use no additional memory to improve convergence with preconditioning.

2. We establish a convergence bound of AdaMeZO under a non-convex assumption that recovers the convergence rate of preconditioned MeZO with multiples of memory cost.

3. We conduct extensive experiments to evaluate AdaMeZO's performance of AdaMeZO. We first employ 2-dimensional toy functions and visualize the trajectories of optimization. They demonstrate that AdaMeZO converges to optimal points, whereas MeZO does not, even with the same step budgets. Then we demonstrate AdaMeZO's performance by fine-tuning different models (RoBERTa (Liu et al., 2019b), OPT (Zhang et al., 2022a), and LLaMa (Touvron et al., 2023)) for a task set identical to MeZO's. It is found that AdaMeZO almost always reaches the same termination condition as MeZO, with up to $70\%$ fewer forward passes and higher performance.

---

[1]Codes are available at `https://github.com/shawnnn3di/AdaMeZO`.

## 2. Related Works

### 2.1. Zeroth-order Optimizers for LLMs

Zeroth-order optimization is also known as derivative-free or black-box optimization. Previously, it was used for situations where the objective function has no derivatives or obtaining derivatives is expensive. Fine-tuning LLMs falls into the latter case and sometimes both for non-differentiable objectives (Tang et al., 2023; Zhang et al., 2024c). In the context of modern deep learning, it translates to the emission of auto-differentiation by backward propagation (Rumelhart et al., 1986), resulting in hugely reduced memory consumption. Some past work on zeroth-order optimizers include (Spall, 1992; 1997; Vakhitov et al., 2009; Agarwal et al., 2009; Raginsky & Rakhlin, 2011; Jamieson et al., 2012; Wang et al., 2020; FairScale authors, 2021). MeZO (Malladi et al., 2023) firstly adopts the classical SPSA (Spall, 1992) to fine-tune billion-level dimension LLMs based on low rank assumptions on LLM fine-tuning, achieving comparable performance with much fewer GPU hours. A survey on concurrent extensions on top of MeZO can be found in Section A. Notably, from-scratch zeroth-order optimization on smaller networks is also of great research interest (Chen et al., 2023).

### 2.2. First-order Optimizers for LLMs

First-order optimization algorithms form the backbone of training or fine-tuning LLMs, offering computational efficiency and scalability across billions of parameters. One of the most classic solutions is Adam (Kingma & Ba, 2014), which updates based on first- and second-order moments. Some of its variants are as follows. AdamW (Loshchilov & Hutter, 2017) introduces adaptive learning rates via moment estimates, achieving faster convergence on nonconvex objectives. LAMB (You et al., 2019) features a layer-wise adaptation strategy to accelerate the training of large models employing large batches. Adafactor (Shazeer & Stern, 2018) reduces memory usage by maintaining factored second-moment estimates rather than the faithful estimates. AdaBelief (Zhuang et al., 2020) replaces Adam's second moment estimates with a squared gradient with the squared difference between the gradient and its running mean to improve convergence and generalization. Lion (Chen et al., 2022) uses only sign-based moment updates without per-parameter scaling to reduce memory costs. Adabound (Luo et al., 2019) stabilizes learning rates between dynamic lower and upper thresholds to transition from adaptive behavior to SGD-like stability. RAdam (Liu et al., 2019a) introduces rectification to stabilize adaptive learning rates, improving training stability in the early iterations. (Defazio et al., 2024) introduced a schedule-free optimizer that requires no additional hyperparameters beyond those of standard optimizers with momentum. In-terestingly, (Zhang et al., 2024b) finds that block structures of diagonal Hessians can help reduce memory costs without harming performance. All of these variants feature empirical estimations of first and second moments, but with changes such as moment centering and regularization, which implies that the proposed method can also be applied to the zeroth-order versions of these variants. Additionally, (Pethick et al., 2025) explores leveraging linear minimization oracles to adapt to loss landscapes without Adam-style updates.

### 2.3. Acceleration by Adam

Compared with first-order optimizers, second-order informed optimizers incorporate second-order information during gradient calculation. The design of Adam mimics Newton's method with second derivatives. Specifically, the second moment can be viewed as a rough approximation to the inverse Hessian. Lines of work provide analytic or numeric support for Adam's near-diagonal Hessians estimation in deep learning. (Das et al., 2024) formalizes that diagonally-dominant Hessians make Adam mathematically faster. (Zhang et al., 2024a) finds block-diagonal Hessians in real neural networks and shows Adam outperforms SGD precisely due to this structure. Empirically, (El-sayed et al., 2024) measures strong diagonal dominance in MLP Hessians. (Gui et al., 2021) demonstrates that over-parameterization further drives the Hessian toward a diagonal form. Interestingly, (Ghorbani et al., 2019) found that the Hessian spectra of deep neural networks become stable after less than $1\%$ training step budget. (Kunstner et al., 2023) finds that Adam's great performance could be attributed to its similarity to sign descent with momenta.

## 3. Methods

In this section, we first introduce the classic forward-pass-only gradient estimator, SPSA, which is the foundation of MeZO (Malladi et al., 2023). Then, we will explain why direct splicing of SPSA with the Adam-style update rule leads to excessive memory usage and how our technique can prevent this.

### 3.1. Preliminaries

**Definition 3.1** (Simultaneous Perturbation Stochastic Approximation, SPSA (Spall, 1992))**.** *Given a model with weight $\boldsymbol{w}_t$ at step $t$ and objective function $\mathcal{L}$, SPSA estimates the gradient on a batch $\mathcal{B}$ with perturbation scale $\mu > 0$ and random direction $\boldsymbol{z}_t$ as*

$$\boldsymbol{g}_t = \frac{\mathcal{L}(\boldsymbol{w}_{t-1} + \mu\boldsymbol{z}_t, \mathcal{B}_t) - \mathcal{L}(\boldsymbol{w}_{t-1} - \mu\boldsymbol{z}_t, \mathcal{B}_t)}{2\mu}\boldsymbol{z}_t. \quad (1)$$

Following prior works, we assume $\boldsymbol{z} \sim \mathcal{N}(\boldsymbol{0}, I_d)$. It can

**Algorithm 1** $h$-MeZO

---

**Input:** Initialized model parameters $\boldsymbol{w}_0 \in \mathbb{R}^d$, loss function $\mathcal{L}: \mathbb{R}^d \to \mathbb{R}$, step budget $T$, perturbation scale $\mu$, learning rate $\eta$, horizon $h$, first EMA ratio $\beta_1$

**Output:** Trained model parameters $\boldsymbol{w}_T$

```
seeds, projs ← [], []
```
**for** $t = 1, \dots, T$ **do**
  Sample batch $\mathcal{B}_t$ and random seed $s$
  Reset the PRNG with random seed $s$, spawn $\boldsymbol{z}_t \sim \mathcal{N}(\boldsymbol{0}, I_d)$
  Estimate $p_t$ using Equation (1)     # *in-place model perturbation*
```
  seeds.append(s), projs.append(p_t)
```
  $\boldsymbol{w}_t \leftarrow \boldsymbol{w}_t$
  **for** $\tau = 1, \dots, h$ **do**
```
    p ← projs[−τ], s ← seeds[−τ]
```
    Reset the PRNG with random seed $s$, spawn $\boldsymbol{z} \sim \mathcal{N}(\boldsymbol{0}, I_d)$
    $\boldsymbol{w}_t \leftarrow \boldsymbol{w}_t - \eta \beta_1^{\tau-1} p \boldsymbol{z}$
  **end for**
**end for**

---

be shown that $\boldsymbol{g}_t \to \nabla\mathcal{L}(\boldsymbol{w}_t, \mathcal{B}_t)$ as $\mu \to 0$, and is treated as an unbiased gradient estimator with a sufficiently small perturbation scale $\mu$. With an SGD-styled update rule of $\boldsymbol{w}_t \leftarrow \boldsymbol{w}_t - \eta\boldsymbol{g}_t$, modifying the model parameters for gradient estimation and model update can be done in-place. MeZO runs quickly on GPUs since they can spawn random gradients the size of 77.5 B within a second[2]. As a result, MeZO generates less information in memory during fine-tuning than backpropagation. The method is shown to yield competitive performance.

### 3.2. First Moment Can Be Recovered Without Additional Memory

Using the first moment computed by history gradients with EMA updates is a widely used technique to cancel out instantaneous gradient noise, thereby promoting convergence. Common first-order algorithms require an additional trunk of memory of size $P$ to store the current first moment $\boldsymbol{m}_t$ as follows:

$$\boldsymbol{m}_t \leftarrow \beta_1\boldsymbol{m}_t + (1-\beta_1)\boldsymbol{g}_t, \quad \boldsymbol{w}_t \leftarrow \boldsymbol{w}_t - \eta\boldsymbol{m}_t.$$

However, the MeZO-style in-place parameter update allows the first moment to be approximated without storing history gradients. Specifically, we unroll the recursion into independent gradients, set a hyperparameter, the horizon $h$, and discard the outdated gradients computed more than $h$ steps ago, then employ a similar in-place parameter update process as in MeZO as Equation (2) and detailed in Algo-

---

[2] https://developer.nvidia.com/curand

rithm 1.

$$\boldsymbol{m}_t \approx (1-\beta_1)(\boldsymbol{g}_t + \beta_1\boldsymbol{g}_{t-1} + \cdots + \beta_1^{t-h-1}\boldsymbol{g}_{t-h-1}). \tag{2}$$

*Remark* 3.2. The idea behind Algorithm 1 is that the share of a history gradient $\boldsymbol{g}_{t-t'}$ decays quickly. As an example, after sufficiently long steps, the share of $\boldsymbol{g}_{t-10}$ approximates $0.9^{10}/(1/(1-0.9)) \approx 0.0387$ at $\beta_1 = 0.9$. It implies that the key components of the first moment are several of the most recent gradients, while the rest are relatively safe to be omitted. Supported by the PRNG as a coder of the random gradients, Algorithm 1 can use truncated first moments without additional memory.

### 3.3. Second Moment Informed Updates Without Additional Memory

We can similarly recover a truncated second moment. However, bringing them into the update will still be memory-intensive. We investigate the issue and present our solution in this subsection.

An Adam-style update rule can be expressed as follows:

$$\begin{aligned} \boldsymbol{m}_t &\leftarrow \beta_1\boldsymbol{m}_t + (1-\beta_1)\boldsymbol{g}_t, \\ \boldsymbol{v}_t &\leftarrow \beta_2\boldsymbol{v}_t + (1-\beta_2)\boldsymbol{g}_t \odot \boldsymbol{g}_t, \\ \boldsymbol{w}_t &\leftarrow \boldsymbol{w}_t - \eta\frac{\boldsymbol{m}_t}{\sqrt{\boldsymbol{v}_t + \epsilon}}. \end{aligned}$$

Unlike Equation (2), we can't decompose the updates into independent gradients. Instead, we will get a summation of preconditioned gradients as follows:

$$\begin{aligned} \frac{\boldsymbol{m}_t}{\sqrt{\boldsymbol{v}_t + \epsilon}} \approx &(1-\beta_1)(\frac{\boldsymbol{g}_t}{\sqrt{\boldsymbol{v}_t + \epsilon}} + \frac{\beta_1\boldsymbol{g}_{t-1}}{\sqrt{\boldsymbol{v}_t + \epsilon}} + \dots \\ &+ \frac{\beta_1^{t-h-1}\boldsymbol{g}_{t-h-1}}{\sqrt{\boldsymbol{v}_t + \epsilon}}), \end{aligned}$$

with a common conditioner $\sqrt{\boldsymbol{v}_t + \epsilon}$ that can only be recovered with multiple gradients as follows:

$$\begin{aligned} \boldsymbol{v}_t \approx &(1-\beta_2)(\boldsymbol{g}_t \odot \boldsymbol{g}_t + \beta_2\boldsymbol{g}_{t-1} \odot \boldsymbol{g}_{t-1} + \dots \\ &+ \beta_2^{t-h-1}\boldsymbol{g}_{t-h-1} \odot \boldsymbol{g}_{t-h-1}). \end{aligned}$$

In other words, it is impossible to perform the update by multiple PRNG overwrites of the model weights. As a result, the preconditioner should be maintained in memory before modifying the model weights, which incurs substantial memory cost, as in previous work such as (Zhao et al., 2024b). We aim to remove this additional memory cost in the following subsections.

#### 3.3.1. FINE-SCALED RANDOM STREAM GENERATION BY STATE CACHING

Since saving the entire preconditioner is memory-intensive, we try to perform blockwise preconditioned updates at a

**Algorithm 2** PRNG

---

**Input:** Seed $s$

**Output:** Random number streams $r_n$

Initial state mapper $I$ maps the random seed $s$ to the initial state $S_0$.

**while** No stop signal **do**

    PRNG outputs random stream $\{r_n\}$ by the recurrence

$$r_n = O(S_{n-1}), \quad S_n = F(S_{n-1}), \qquad (3)$$

**end while**

---

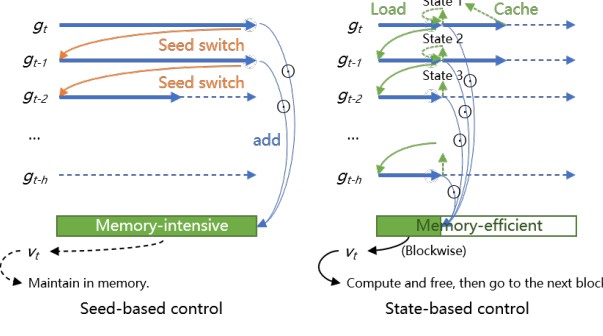

*Figure 2.* Block-wise moment approximation in AdaMeZO. $\odot$ denotes the Hadamard product.

lower additional memory cost, which is not available in normal use of PRNG. A formal expression of how concurrent PRNG algorithms generate random number streams is as Algorithm 2, with algorithm-specified and deterministic state update function $F$ and extractor $O$.

For prior practices including (Malladi et al., 2023; Zhao et al., 2024a;b), the above process guarantees identical and complete $z_t$ for gradient estimation, and gradient updating is obtained by caching the seed $s$, so that in-place gradient estimation and parameter updating without additional memory access functions. However, caching the random state $S$ offers the possibility to *jump* to a specified position in a random number stream. It allows the PRNG to faithfully continue a particular random stream from wherever it left off, making it more flexible than seed caching. Code examples for this feature can be found in Section C.

An illustration of the block-wise moment approximation is shown in Figure 2. A parameter block partition $\boldsymbol{w} = \{\boldsymbol{w}^{(1)}, \boldsymbol{w}^{(2)}, \dots, \boldsymbol{w}^{(b)}\}$ is prepared at the beginning of a parameter update. We first start at the first block. Processing of the first block is the same as how prior works exploit PRNGs. However, after the first random direction block $\boldsymbol{z}_{t-t'}^{(1)}$ is spawned, AdaMeZO records $S_n$, the corresponding random state, for each seed. When the spawning of the first block from each of the seeds within the horizon is finished, AdaMeZO skips the initial state mapping in Algorithm 2, and loads the cached $S_n$ to the PRNG for the next block, so that the contiguous random stream is generated rather than starting over again from the first output of the random stream. The process loops until all blocks have finished their updates.

### 3.3.2. ADAM-STYLE UPDATES WITH ZEROTH-ORDER GRADIENTS

With random state caching, we can update models according to Equation (3) block-wise, which is impossible for seed caching as employed by previous works, since seed caching only allows the random stream to be spawned from the initial digit. However, we need some warm-up steps to accumulate history gradients before estimating finite-

horizon moments. Due to page limits, we elaborate on the process in Algorithm 3 of the Appendix.

*Remark* 3.3. It is worth mentioning that caching random states adds a minimal memory cost compared to caching seeds. In Philox (Salmon et al., 2011), the default choice of CUDA PRNG, random state $S$ consists of a 64-bit random seed, a 64-bit subsequence identifier, and a 64-bit offset. The Mersenne Twister (Matsumoto & Nishimura, 1998), the default CPU PRNG, maintains similar information to random states. Therefore, caching the random states incurs a negligible additional memory cost at the bit level compared to caching seeds.

*Remark* 3.4. Though the first and second moments do not go into the memory, recovering them requires a temporary additional memory trunk, whose size scales to the size of the block, corresponding to the $\delta P$ term in Table 1. A natural block strategy is the different layers of the model. If the model consists of 32 layers, the additional memory introduced in the second moment approximation is roughly $2/32P$ ($1/32P$ for $\boldsymbol{m}_t^{(b)}$ and $\boldsymbol{v}_t^{(b)}$ each). However, the block can be made as small as a single 1-parameter block. Therefore, AdaMeZO can theoretically approximate the moments by performing frequent random state dumps and loads without incurring additional memory requirements.

## 4. Theory

We employ the following widely adopted assumptions to facilitate an analysis.

**Assumption 4.1** (L-smooth). *For any weight vector $\boldsymbol{w}_1, \boldsymbol{w}_2 \in \mathbb{R}^d$, for a constant $0 < L < \infty$ it holds that*

$$\mathcal{L}(\boldsymbol{w}_2) \leq \mathcal{L}(\boldsymbol{w}_1) + \langle \nabla \mathcal{L}(\boldsymbol{w}_1), \boldsymbol{w}_2 - \boldsymbol{w}_1 \rangle + \frac{L}{2}\|\boldsymbol{w}_2 - \boldsymbol{w}_1\|_2^2.$$

**Assumption 4.2** (Bounded gradient variance). *The stochastic gradient $\nabla \mathcal{L}(\boldsymbol{w}_t, \mathcal{B}_t)$ has no bias and $\sigma^2$ vari-*

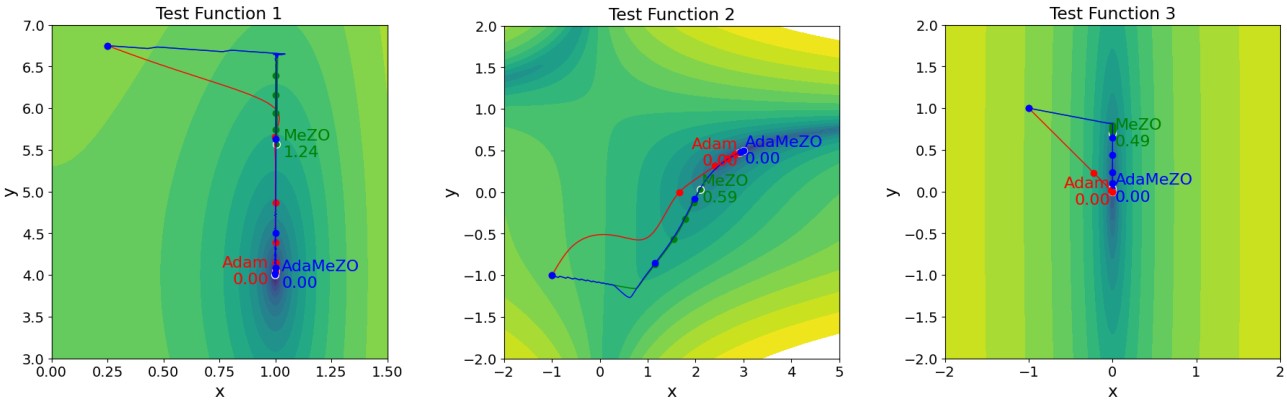

*Figure 3.* Optimization trajectories on test functions. The loss values at termination are labeled.

*ance due to batch stochasticity, specifically*

$$\mathbb{E}_{\mathcal{B}_t}[\nabla \mathcal{L}(\boldsymbol{w}_t, \mathcal{B}_t)] - \nabla \mathcal{L}(\boldsymbol{w}_t) = 0, \qquad (4)$$

$$\mathbb{E}_{\mathcal{B}_t}[\|\nabla \mathcal{L}(\boldsymbol{w}_t, \mathcal{B}_t)\|_2^2] - \|\nabla \mathcal{L}(\boldsymbol{w}_t)\|_2^2 \leq \sigma_t^2, \quad \sigma_t < \infty.$$

**Assumption 4.3** (Bounded second moment, (Zhao et al., 2024b)). *Each entry of $\Sigma_t$ lies in the range $[s_l, s_u]$ with $0 < s_l < s_u < \infty$.*

**Assumption 4.4** (Finite gradient drift within horizon). *The gradient drift within the moment horizon $h$ is finite, specifically, $\boldsymbol{m}_t$ as the first moment at step $t$ satisfies*

$$\|\boldsymbol{m}_t - \nabla \mathcal{L}(\boldsymbol{w}_t)\|_2 \leq \mathcal{O}((1 - \beta_1)L\eta).$$

**Lemma 4.5** ((Magnus et al., 1978)). *Let $A$ and $B$ be two symmetric matrices, $\boldsymbol{z} \sim \mathcal{N}(\boldsymbol{0}, \boldsymbol{I}_d)$. Define $\boldsymbol{x} = \boldsymbol{z}^\top A \boldsymbol{z} \boldsymbol{z}^\top B \boldsymbol{z}$, then it holds that*

$$\mathbb{E}_{\boldsymbol{z}}[\boldsymbol{x}] = (\mathrm{tr}A)(\mathrm{tr}B) + 2\mathrm{tr}(AB).$$

**Assumption 4.6** (Local $r$-effective rank, (Malladi et al., 2023)). *Let $G(\boldsymbol{w}_t) := \max_{\mathcal{B}, |\mathcal{B}|=1} \|\nabla \mathcal{L}(\boldsymbol{w}_t, \mathcal{B})\|$. There is a matrix $\mathcal{H}(\boldsymbol{w}_t) \preceq LI_d$ satisfying:*

1. *For all $\boldsymbol{w}$ such that $\|\boldsymbol{w} - \boldsymbol{w}_t\|_2 \leq \eta d G(\boldsymbol{w}_t)$, it holds that $\nabla^2 \mathcal{L}(\boldsymbol{w}) \preceq H(\boldsymbol{w}_t)$.*

2. *The effective rank of $H(\boldsymbol{w}_t)$, specifically, $\mathrm{tr}(\mathcal{H}(\boldsymbol{w}_t))/\|\mathcal{H}(\boldsymbol{w}_t)\|_{op}$, is at most $r$.*

We present a convergence bound for non-convex optimization.

**Theorem 4.7.** *With a sufficiently small learning rate $\eta$, AdaMeZO converges to a stationary point with*

$$\mathbb{E}\left[\frac{1}{T}\sum_{t=1}^{T}\|\nabla \mathcal{L}(\boldsymbol{w}_t)\|_2^2\right] \leq \mathcal{O}\left(\frac{1}{\sqrt{T}}\right) + \mathcal{O}(\mu^2).$$

Detailed proof can be found in Appendix E. The bound recovers the structure from (Zhao et al., 2024b). The above result shows that after $T = \mathcal{O}(\epsilon^{-2})$ steps, AdaMeZO converges to a small neighborhood of a stationary point satisfying $\mathbb{E}\left[\|\frac{1}{T}\sum_{t=1}^{T}\nabla \mathcal{L}(\boldsymbol{w}_t)\|_2^2\right] < \epsilon$.

## 5. Experiment Results

We present empirical results for AdaMeZO with its baselines in this section. Generally, there are two types of LLMs: 1) encoder-decoder, or masked language models (MLM), such as BERT (Devlin et al., 2019) and its variants, and 2) decoder-only, or autoregressive models (ARM), such as GPT, OPT, and LLaMA families. To comprehensively demonstrate AdaMeZO's performance, we first illustrate the optimization trajectories for toy functions. Then, we test AdaMeZO with baseline algorithms on well-recognized LLMs, including an MLM RoBERTa (Liu et al., 2019b), and two ARMs, OPT (Zhang et al., 2022a) and LlaMa(Touvron et al., 2023).

### 5.1. Toy Functions

It is impractical to visualize trajectories of model optimization with billions of dimensions. However, we can illustrate the optimization trajectories on three 2-dimensional toy functions as in Figure 3 to show how AdaMeZO adapts to heterogeneous curvatures. We test the Adam optimizer (Kingma & Ba, 2014) implemented in PyTorch (Paszke, 2019), the vanilla MeZO (Malladi et al., 2023), and the proposed AdaMeZO. More details in Section B.5.

In general, we observe that AdaMeZO shares Adam's curvature adaptability. Although AdaMeZO walks longer paths due to stochastic gradient directions and warm-up steps, it moves swiftly in regions of low curvature thanks to the preconditioning provided by the diagonal Hessian

*Table 2.* Results on RoBERTa-large over language tasks with $k = 16$.

| Task Type | SST-2 | SST-5 | SNLI | MNLI | RTE | TREC | Average |
| --- | --- | --- | --- | --- | --- | --- | --- |
| | —- sentiment —- | | - natural language inference - | | | – topic – | |
| Zero-shot | 79.0 | 35.5 | 50.2 | 48.8 | 51.4 | 32.0 | 49.4 |
| FO ($\geq 4\times$ memory) | 91.8 | 47.5 | 77.5 | 70.0 | 66.4 | 85.0 | 73.0 |
| MeZO | 90.6 | 44.1 | **67.3** | 58.1 | 61.6 | 67.3 | 64.8 |
| | (1.4) | (1.0) | (3.1) | (1.1) | (1.3) | (2.7) | – |
| MeZO-switch | 90.6 | 44.3 | **67.3** | 58.0 | 61.6 | 67.0 | 64.8 |
| | (1.6) | (1.6) | (2.8) | (1.3) | (2.0) | (4.7) | – |
| *AdaMeZO* | **90.9** | **45.2** | 66.8 | **58.6** | **63.1** | **71.5** | **66.0** |
| | (0.9) | (2.0) | (2.9) | (1.4) | (2.3) | (5.2) | – |

*Table 3.* Main results on OPT-1.3B over language tasks. Avg (w.o S,D) indicates the average metric except SQuAD and DROP.

| Task Type | SST-2 | RTE | CB | BoolQ | WSC | WIC | MultiRC | COPA | ReCoRD | SQuAD | DROP | Avg | Avg (w.o S,D) |
| --- | --- | --- | --- | --- | --- | --- | --- | --- | --- | --- | --- | --- | --- |
| | | | —————— classification —————— | | | | | – multiple choice – | | —— generation —— | | | |
| Zero-shot | 53.5 | 53.4 | 39.2 | 45.5 | 43.2 | 57.5 | 45.4 | 75.0 | 70.5 | 27.2 | 11.1 | 47.4 | 53.6 |
| FO ($\geq 4\times$ memory) | 90.9 | 64.0 | 77.2 | 64.4 | 52.8 | 62.3 | 65.2 | 74.0 | 69.1 | 80.4 | 28.2 | 66.2 | 68.9 |
| | (1.2) | (10.7) | (7.9) | (9.3) | (2.0) | (1.9) | (6.0) | (2.9) | (1.2) | (1.5) | (1.7) | – | – |
| MeZO | 90.9 | 52.5 | 65.5 | 61.8 | 51.1 | **58.6** | 53.7 | 74.5 | 70.6 | 73.3 | 22.8 | 61.4 | 64.4 |
| | (0.3) | (1.5) | (6.9) | (2.1) | (8.4) | (1.4) | (2.2) | (3.6) | (1.0) | (0.2) | (0.6) | – | – |
| MeZO-switch | 91.0 | 53.8 | 68.7 | 61.9 | 52.1 | 58.3 | 54.9 | **75.5** | 71.0 | 73.7 | 24.3 | 62.3 | 65.2 |
| | (0.6) | (1.6) | (2.3) | (0.6) | (7.6) | (1.6) | (1.5) | (3.6) | (1.2) | (1.2) | (1.3) | – | – |
| HiZOO | 90.9 | **54.5** | 63.3 | 62.7 | 49.4 | 58.4 | 55.4 | 74.0 | 70.8 | 74.5 | 24.5 | 61.7 | 64.4 |
| | (1.0) | (1.6) | (8.5) | (1.6) | (6.9) | (0.4) | (1.7) | (1.8) | (0.8) | (0.4) | (0.5) | – | – |
| *AdaMeZO* | **91.6** | 54.3 | **69.6** | **63.2** | **53.5** | 58.4 | **55.9** | **75.5** | **71.1** | **76.1** | 24.6 | **63.1** | **65.9** |
| | (0.3) | (3.1) | (1.4) | (1.6) | (7.8) | (1.6) | (0.7) | (4.0) | (1.3) | (0.7) | (1.0) | – | – |

*Table 4.* Memory profile (MB) on standard PyTorch build, measured on OPT-1.3b, batch size=1. Measured via `nvidia-smi`.

| Optimizer | SST2 | COPA | SQuAD | Memory |
| --- | --- | --- | --- | --- |
| MeZO | 5016 | 5058 | 5040 | 1x |
| Adam | 22172 | 21660 | 22688 | 4.40x |
| HiZOO | 7532 | 7535 | 7396 | 1.49x |
| *AdaMeZO* | **5410** | **5452** | **5434** | **1.07x** |

estimator, and the final loss values are comparable to those of Adam. In contrast, MeZO struggles with oscillations in low-curvature regions, leading to worse convergence.

## 5.2. Main Results

We compare the performance of AdaMeZO with vanilla MeZO and MeZO-switch, a variant of MeZO in which the learning rate is manually adjusted to ensure its optimization trajectory is longer than that of AdaMeZO. This ensures that AdaMeZO's outperformance is not due to MeZO's underfitting, but rather to its adaptability to the loss landscape. Additionally, we include HiZOO (Zhao et al., 2024b) as a strong baseline that maintains preconditioners in memory. We set $h = 10, \beta_1 = 0.7, \beta_2 = 0.9$ to evaluate performance across 4 randomly sampled data subsets of the same size and report the mean and standard deviation of the corresponding metric for each task after a hyperparameter study presented in Appendix B.3.

Consistent with previous research (Malladi et al., 2023), we conduct experiments on RoBERTa-large 350M on three types of NLP tasks: sentiment, natural language inference, and topic. We sample $k = 16$ examples per class to demonstrate training performance in a few-shot scenario (see Table 2). It is found that:

**AdaMeZO yields better performance.** Averaged across all tasks, AdaMeZO achieves a **1.2%** absolute accuracy improvement to MeZO on average, with particularly strong gains in tasks like RTE (**1.5%**), TREC (**4.2%**).

Then we extend our investigation to two autoregressive architectures: OPT (Table 3) and LLaMA3 (Table 5). Experimental results show that:

**AdaMeZO's superior performance scales up to billion-level LLMs.** On OPT-1.3B, AdaMeZO surpasses MeZO and MeZO-switch in all but one task. AdaMeZO achieves a **1.7%** absolute accuracy improvement to MeZO on average, with particularly strong gains in tasks like CB (**4.1%**), SQuAD (**2.8%**), WSC (**2.4%**). AdaMeZO also outperforms HiZOO with an average lead of **1.5%**. For LLaMA-3B, AdaMeZO further extends its lead, achieves a **2.7%** absolute accuracy improvement to MeZO on average, with particularly strong gains in tasks like SST2 (**8.1%**), MultiRC (**4.3%**), WSC (**4.1%**), COPA (**4.0%**). AdaMeZO demonstrates its scalability and maintains its advantage in modern-scale models like OPT-30B, as reported in Table 6.

*Table 5.* Main results on LLaMA3-3B over language tasks.

| Task Type | SST-2 | RTE | CB | BoolQ | WSC | WIC | MultiRC | COPA | ReCoRD | SQuAD | DROP | Avg | Avg (w.o S,D) |
|---|---|---|---|---|---|---|---|---|---|---|---|---|---|
| | — classification — | | | | | | | — multiple choice — | | — generation — | | | |
| Zero-shot | 56.0 | 52.7 | 51.6 | 60.9 | 36.5 | 54.3 | 44.8 | 75.0 | 68.2 | 47.3 | 20.8 | 51.6 | 55.5 |
| FO ($\geq 4\times$ memory) | 92.5 | 73.9 | 85.6 | 65.9 | 57.8 | 67.1 | 70.6 | 75.7 | 68.6 | 83.9 | 32.2 | 70.3 | 73.1 |
| | (0.7) | (5.4) | (6.9) | (7.3) | (7.6) | (0.7) | (1.8) | (2.6) | (1.0) | (0.3) | (1.8) | – | – |
| MeZO | 84.5 | 53.2 | 64.7 | 62.6 | 50.4 | 54.6 | 52.6 | 77.2 | 70.0 | 79.2 | 26.8 | 61.4 | 63.3 |
| | (4.9) | (0.7) | (2.6) | (0.7) | (11.3) | (0.3) | (2.5) | (2.0) | (0.4) | (0.9) | (0.5) | – | – |
| MeZO-switch | 86.6 | 54.1 | 65.5 | 63.2 | 51.6 | 54.7 | 54.7 | 78.7 | 70.4 | **80.4** | 27.6 | 62.5 | 64.4 |
| | (4.5) | (1.5) | (0.9) | (0.3) | (12.2) | (1.0) | (0.6) | (2.2) | (0.6) | (0.9) | (0.6) | – | – |
| HiZOO | 92.2 | 54.1 | 65.1 | 63.7 | 52.8 | **54.9** | 56.5 | **82.5** | **71.5** | 18.5 | 6.1 | 56.1 | 65.9 |
| | (0.5) | (0.3) | (2.2) | (0.3) | (5.4) | (1.7) | (0.7) | (0.5) | (0.6) | (1.9) | (1.2) | – | – |
| ***AdaMeZO*** | **92.6** | **54.4** | **66.0** | **64.6** | **54.5** | **54.9** | **56.9** | 81.2 | 71.3 | **80.4** | **28.1** | **64.1** | **66.3** |
| | (0.5) | (1.5) | (1.4) | (2.6) | (7.5) | (1.6) | (1.0) | (3.2) | (0.9) | (1.8) | (1.1) | – | – |

*Table 6.* Main results on OPT-30B over language tasks, with prefix-tuning.

| Task | SST-2 | WSC | WIC | COPA | Avg |
|---|---|---|---|---|---|
| Zero-shot | 56.6 | 38.4 | 50.1 | 81.0 | 56.5 |
| HiZOO | 90.1 | 56.9 | 55.2 | 86.2 | 72.1 |
| | (1.1) | (6.2) | (3.6) | (1.7) | – |
| ***AdaMeZO*** | **91.1** | **57.6** | **57.3** | **87.0** | **73.2** |
| | (0.6) | (2.4) | (1.5) | (1.4) | – |

*Table 7.* Runtime profile (sec/step) on standard PyTorch build, measured on OPT-1.3b, batch size=1.

| Optimizer | SST2 | COPA | SQuAD |
|---|---|---|---|
| MeZO | 0.21 | 0.18 | 0.21 |
| HiZOO | 0.23 | 0.24 | 0.25 |
| MeZO + a. | 0.23 | 0.23 | 0.24 |
| MeZO + a. + b. (AdaMeZO) | 0.31 | 0.30 | 0.31 |
| Adam | 0.12 | 0.13 | 0.13 |

We defer results of larger scale to Appendix D.1.

### 5.3. Memory Efficiency

AdaMeZO incurs a small additional memory due to block-wise moment caching as reported in Table 4. We can observe that, compared to optimizers that maintain actual moments, the additional memory cost is significantly reduced.

### 5.4. Wall-clock Time Analysis

AdaMeZO incurs longer per-step runtime compared to MeZO, mainly due to a) the additional PRNG calls for past gradient regeneration, and b) the weighted gradient accumulation for moment recovery. We report a runtime profile as Table 7. We observe that the main contributor to AdaMeZO's additional runtime is the accumulation of past gradients that are regenerated. Optimizing this accumulation process or using prefix tuning will narrow the speed gap relative to MeZO.

## 6. Conclusion, Limitations and Future Works

In this work, we introduce AdaMeZO, the first ZO optimizer that incorporates Adam-style first- and second-moment updates without doubling or tripling the memory requirements of the original MeZO. This is achieved by estimating truncated moments and performing more refined operations on PRNGs. We provide theoretical analysis and empirical evaluations. Visualizations show that AdaMeZO adapts to complex loss landscapes without consuming excessive additional memory. Experiments on well-recognized models show that AdaMeZO reaches on-par performance using fewer forward passes and can continue to lower loss values before reaching identical terminal conditions. The paper's limitations are as follows.

We have captured the gradient drift across different steps using a big $\mathcal{O}$ constant related to L-smoothness, EMA weight $\beta_1$, and learning rate. Although finite moment horizons may help to keep the estimations less biased, we did not attempt to explicitly capture the gap, which is a future research direction.

AdaMeZO estimates second moments at a small cost, but they are inaccurate. The reason is two-fold: 1) AdaMeZO runs on zeroth-order gradient estimations, and 2) a smaller $\beta_2$ to guarantee that the discarded part contributes only a small share. Future investigations into more accurate second-moment estimations could improve performance.

## Impact Statement

This paper presents work whose goal is to advance the field of Machine Learning. There are many potential societal consequences of our work, none which we feel must be specifically highlighted here.

## Acknowledgments

This work is supported in part by the National Natural Science Foundation of China (Grant No. 62522118,

62371313), in part by the Shenzhen Science and Technology Program (Grant No. JCYJ20241202124934046), in part by the Guangdong Young Talent Research Project (Grant No. 2023TQ07A708), and in part by Shenzhen Loop Area Institute (Contract No. SLAI2026020007).

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

# A. Additional Related Works

In addition to MeZO, numerous subsequent excellent works have emerged to enhance the vanilla version. (Jiang et al., 2024) incorporates uncertain moments estimations to promote convergence. (Zhao et al., 2024a) invokes Adam-style update rules for better performance. (Zhao et al., 2024b) estimates the diagonal Hessian with a three-point second derivative estimation, admitting a third forward pass for each step. (Liu et al., 2024; Guo et al., 2024) proposed to insert sparsity for better performance. (Chen et al., 2024; Sun et al., 2025) exploits the low-rank property for better performance. (Chen et al., 2025a) proposes a hybrid optimizer that balances efficiency and trade-offs. (Tan et al., 2025) explores a layer-wise adaptation to speed up zeroth-order fine-tuning. (Chen et al., 2025b) investigates memory-efficient zeroth-order fine-tuning from a subspace-optimization perspective. (Yu et al.) introduces a block version of HiZOO, attempting to preserve preconditioning-improved convergence while reducing additional memory access.

# B. Detailed Experiment Settings

## B.1. Computation Resources

We summarize the computational devices for empirical evaluations in Table 8. We use device 1 for MLM experiments and device 2 for ARM experiments.

*Table 8.* Summary of computational devices for empirical evaluations.

| Device | OS/CPU/GPU | Python | PyTorch | CUDA | cuDNN |
|---|---|---|---|---|---|
| 1 | Linux 5.10.0, amd64
Intel(R) Xeon(R) Gold 6133 CPU @ 2.50GHz
6x NVIDIA GeForce RTX 3090 GPU | 3.10.13 | 2.3.0 | 12.1 | 8.9 |
| 2 | Linux 4.18.0, x86_64
AMD EPYC 7742 64-Core Processor
4x NVIDIA A100-SXM4-80GB | 3.9.7 | 2.1.0 | 12.1 | 8.9 |

## B.2. Formal Pseudo-codes for AdaMeZO

A formal description of AdaMeZO in pseudo-code is as Algorithm 3.

## B.3. Hyperparameter Study

There are 3 new hyperparameters introduced in AdaMeZO, the horizon $h$, and the EMA weight in momentum update $(\beta_1, \beta_2)$ as in Adam.

We first analyze how different $h$ impacts the evaluation loss curve, as shown in Figure 4. We find that setting $h > 5$ admits comparable results, while $h < 5$ requires further optimization to avoid crash due to numerical instability. We set $h = 10$ in experiments for stability, and set $h = 5$ for wall-clock time analysis.

Similar to Adam, two new hyperparameters $(\beta_1, \beta_2)$ are introduced into the algorithm. We report AdaMeZO's performance across different hyperparameter settings, as shown in Table 9, on OPT-1.3B with the SST2 task, since complete task sweeps are computationally expensive. It can be observed that the first moments improve MeZO's performance, and the second moments further improve it. The performance gain is robust against reasonable choices of the hyperparameter $(\beta_1, \beta_2)$.

*Table 9.* Performance comparison with different $(\beta_1, \beta_2)$.

| $(\beta_1, \beta_2)$ | SST2 | COPA | SQuAD |
|---|---|---|---|
| (0.7, 0.9) | 91.6 (0.3) | 75.5 (4.0) | 76.1 (0.7) |
| (0.7, 0.99) | 90.9 (0.9) | 75.3 (2.9) | 75.6 (0.9) |
| (0.6, 0.9) | 91.1 (0.6) | 75.8 (2.9) | 75.6 (1.2) |
| (0.8, 0.9) | 91.5 (0.7) | 74.3 (2.9) | 75.6 (1.5) |
| (0.7, 0.0), mSGD | 90.9 (0.9) | 75.3 (2.9) | 75.6 (0.9) |
| (0.0, 0.0), MeZO | 90.9 (0.3) | 74.5 (3.6) | 73.3 (0.2) |

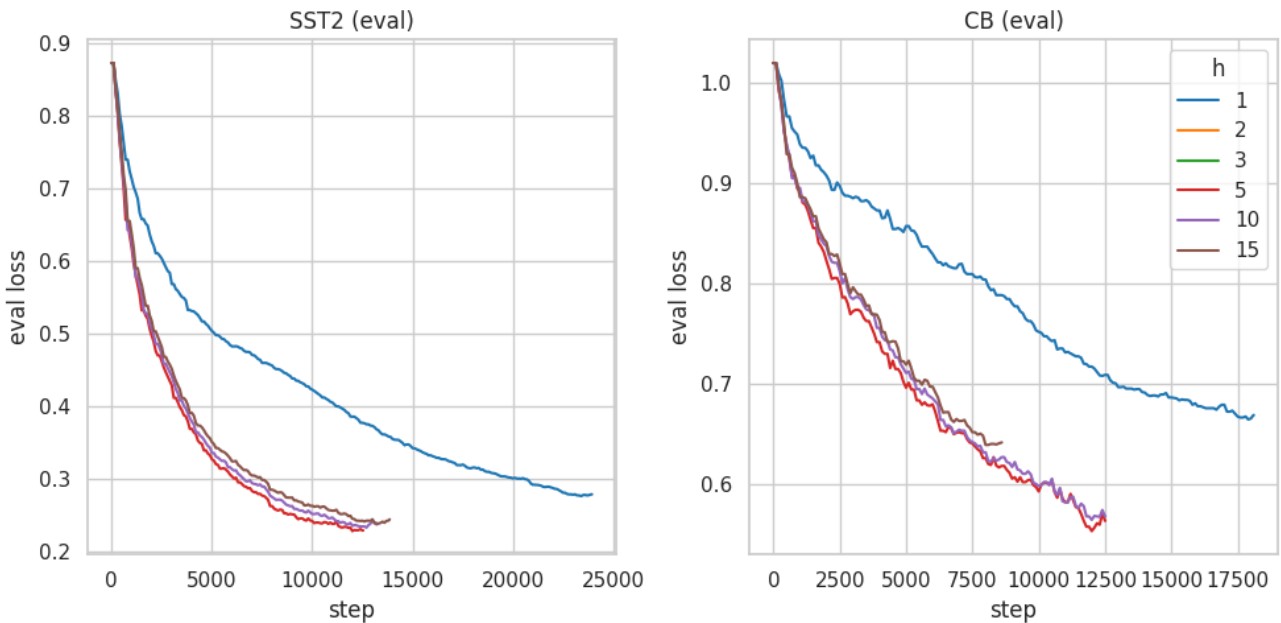

*Figure 4.* Evaluation loss with different $h$.

### B.4. Detailed Settings for Figure 1

For (a), the learning rate is 1e-6, with 16 training samples per class. For (b) and (c), the learning rate is 1e-7, with 1000 training samples in total. We set $\beta_1 = 0.7, \beta_2 = 0.9, h = 10$, so that AdaMeZO discards only a small part by truncating the moments and admits a smoother second moment estimation compared to the first. With an abuse of context, the choice of $(\beta_1, \beta_2)$ falls into the suggested area $0 < \beta_1 < \sqrt{\beta_2} < 1$ by (Zhang et al., 2022b). Fine-tuning terminates when either of the following happens.

1. Measure evaluation loss per 100 steps. Evaluation loss does not drop for 5 continual measures.

2. Number of steps exceeds 40000.

### B.5. Detailed Settings for Section 5.1

The expressions of the test functions are

1. $f_1(x, y) = 8(x - 1)^2(1.3x^2 + 2x + 1) + 0.5(y - 4)^2$ (Zhao et al., 2024b).

2. $f_2(x, y) = (1.5 - x + xy)^2 + (2.25 - x + xy^2)^2 + (2.625 - x + xy^3)^2$ (Beale, 1958).

3. $f_3(x, y) = 100x^2 + y^2$.

Specifications on implementations are as Table 10. The setting follows on the following rule:

1. Set the learning rate of Adam to 0.01,

2. Tune the learning rate for ZO optimizers so that the trajectory lengths are comparable to Adam's. We allow a longer trajectory ($< 1.6\times$) for ZO optimizers.

For MeZO and AdaMeZO, we allow only 2 seeds coding 2 gradient directions. This is to capture the situation where the number of steps, equivalently the total number of explored gradient directions (in thousands), is usually less than the number of dimensions of the LLMs (in billions).

Trajectories at higher resolutions and 3D views of the loss landscapes are shown in Figure 5.

*Table 10.* Specifications for toy functions.

| | Adam | | MeZO | | AdaMeZO | | # steps | Initialization |
|---|---|---|---|---|---|---|---|---|
| | lr | length | lr | length | lr | length | | |
| $f_1$ | 0.01 | 3.0227 | 0.01 | 4.6659 | 0.01 | 4.5078 | 600 | $(0.2, 6.75)$ |
| $f_2$ | 0.01 | 4.3597 | 0.002 | 5.5405 | 0.002 | 5.3207 | 2500 | $(-1, -1)$ |
| $f_3$ | 0.01 | 1.4142 | 0.01 | 1.4243 | 0.01 | 1.8577 | 500 | $(-1, 1)$ |

### B.6. Detailed Settings for Section 5.2

*Table 11.* Hyperparameter settings.

| | $B$ | $T$ | $\eta$ | $q$ | $\mu$ | $(\beta_1, \beta_2)$ |
|---|---|---|---|---|---|---|
| Table 2 | 16 | $1 \times 10^5$ | $1 \times 10^{-6}$ | 5 | $1 \times 10^{-3}$ | $(0.7, 0.9)$ |
| Table 3 | 16 | $4 \times 10^4$ | $1 \times 10^{-7}$ | 5 | $1 \times 10^{-3}$ | $(0.7, 0.9)$ |
| Table 5 | 16 | $4 \times 10^4$ | $1 \times 10^{-7}$ | 5 | $1 \times 10^{-3}$ | $(0.7, 0.9)$ |

Fine-tuning terminates when either of the following conditions is met.

1. Measure evaluation loss per 100 steps. Evaluation loss does not drop for $q$ continual measures.

2. Number of steps exceeds $T$.

## C. Code Snippets for Block-wise Gradient Generation

Previous works call PRNG by the following codes.

```
torch.manual_seed(seed)
z = torch.normal(
    mean=0,
    std=1,
    size=param.data.size(),
    dtype=param.data.dtype,
    device=param.device,
)
```

In this work, for block-wise gradient generation, we wish the PRNG to skip the random stream belonging to prior blocks. Therefore, we directly feed random states into the PRNG, thereby skipping the initialization step implied by `manual_seed(seed)`. A snippet to realize the feature is as follows.

```
self.g = torch.Generator(device='cuda')
self.g.set_state(state)      # scheduled state
z = torch.normal(
    mean=0,
    std=1,
    size=param.data.size(),
    dtype=param.data.dtype,
    device=param.device,
    generator=self.g,
)
state = self.g.get_state(state)
```

## D. Additional Experiment Results

### D.1. Larger Models

We report the performance of AdaMeZO on larger models to demonstrate the scalability of the optimizer as Table 12 and Table 13.

We also report the hyperparameter settings in Section D.1 for Table 6.

We include training loss and evaluation loss curves in Figure 6 and Figure 7, respectively.

*Table 12.* Main results on LLaMA-7B over language tasks.

| Task | SST-2 | RTE | CB | BoolQ | WSC | WIC | MultiRC | COPA | ReCoRD | SQuAD | DROP | Avg | Avg (w.o S,D) |
|---|---|---|---|---|---|---|---|---|---|---|---|---|---|
| Type | | | — classification — | | | | | – multiple choice – | | — generation — | | | |
| Zero-shot | 59.7 | 49.8 | 48.2 | 65.0 | 56.7 | 50.6 | 50.5 | 84.0 | 79.9 | 58.6 | 17.5 | 56.4 | 60.4 |
| FO ($\geq 4\times$ memory) | 95.0 | 86.0 | 94.1 | 83.1 | 54.5 | 66.2 | 79.3 | 81.2 | 75.4 | 89.2 | 39.7 | 76.7 | 79.4 |
| | (0.5) | (2.2) | (1.7) | (0.5) | (5.8) | (4.9) | (3.0) | (2.2) | (2.3) | (1.0) | (1.0) | – | – |
| MeZO | 85.7 | 54.7 | 58.8 | 68.3 | 58.1 | 56.9 | 60.9 | 82.5 | 78.0 | 71.9 | 30.9 | 64.2 | 67.1 |
| | (1.9) | (0.5) | (3.8) | (1.5) | (2.9) | (1.7) | (2.7) | (1.2) | (1.8) | (4.5) | (1.1) | – | – |
| MeZO-switch | 87.2 | 55.2 | 60.6 | 68.7 | 60.2 | 56.8 | 60.5 | 84.0 | 80.3 | 78.8 | 32.3 | 65.8 | 68.1 |
| | (0.7) | (1.2) | (6.3) | (1.2) | (1.2) | (0.5) | (2.3) | (0.8) | (0.5) | (3.2) | (1.1) | – | – |
| HiZOO | 90.9 | 59.7 | **63.3** | 70.3 | 59.8 | 57.4 | **62.7** | 83.7 | 79.3 | 21.3 | 4.6 | 59.4 | 69.7 |
| | (2.5) | (2.9) | (0.9) | (1.5) | (6.7) | (0.2) | (2.4) | (1.2) | (1.6) | (1.1) | (0.9) | – | – |
| ***AdaMeZO*** | **91.4** | **61.2** | 62.9 | **70.9** | **60.5** | **57.6** | 62.1 | **84.5** | **80.5** | **84.9** | **36.2** | **68.4** | **70.2** |
| | (2.5) | (2.6) | (1.6) | (2.2) | (2.0) | (1.1) | (2.6) | (3.1) | (0.9) | (0.9) | (2.1) | – | – |

*Table 13.* Main results on OPT-13B over language tasks. OOM indicates that HiZOO encountered an out-of-memory error. A cell is marked as OOM if any of the evaluation seeds trigger an OOM. The official HiZOO implementation only supports single-GPU training, and the memory overhead of some instances exceeds the capacity of our largest GPU (A100 80GB), leading to OOM failures.

| Task | SST-2 | RTE | CB | BoolQ | WSC | WIC | MultiRC | COPA | ReCoRD | SQuAD | DROP | Avg | Avg (w.o S,D) |
|---|---|---|---|---|---|---|---|---|---|---|---|---|---|
| Type | | | — classification — | | | | | – multiple choice – | | — generation — | | | |
| Zero-shot | 58.8 | 59.6 | 46.4 | 59.0 | 38.5 | 55.0 | 46.9 | 80.0 | 81.2 | 46.2 | 14.6 | 53.2 | 58.3 |
| FO ($\geq 4\times$ memory) | 92.0 | 70.8 | 83.9 | 77.1 | 63.5 | 55.0 | 71.1 | 79.0 | 74.1 | 84.9 | 31.3 | 71.1 | 74.0 |
| MeZO | 92.1 | 60.4 | 67.8 | 65.5 | 56.6 | 54.9 | 56.7 | **87.0** | 80.2 | 82.1 | 30.6 | 66.7 | 69.0 |
| | (0.5) | (0.6) | (1.4) | (3.0) | (7.9) | (1.7) | (0.8) | (1.1) | (1.0) | (1.3) | (1.5) | – | – |
| MeZO-switch | 92.6 | 61.6 | 66.9 | 66.2 | 56.9 | 55.4 | 57.5 | 86.0 | **80.5** | 83.4 | 30.5 | 67.0 | 69.3 |
| | (0.2) | (2.5) | (1.0) | (3.7) | (8.4) | (0.7) | (0.5) | (2.7) | (1.1) | (0.8) | (0.9) | – | – |
| HiZOO | 91.5 | 62.5 | **68.2** | OOM | 56.4 | 55.4 | 57.5 | 86.2 | 80.0 | 83.5 | OOM | – | – |
| | (1.3) | (3.7) | (2.2) | – | (8.3) | (1.4) | (0.6) | (0.9) | (1.4) | (1.2) | – | – | – |
| ***AdaMeZO*** | **92.7** | **63.0** | 67.8 | **70.6** | **58.4** | **55.8** | **58.3** | **87.0** | 80.1 | **83.7** | **31.0** | **68.0** | **70.4** |
| | (0.5) | (6.1) | (2.5) | (3.6) | (7.5) | (0.6) | (0.3) | (1.1) | (0.7) | (1.3) | (0.9) | – | – |

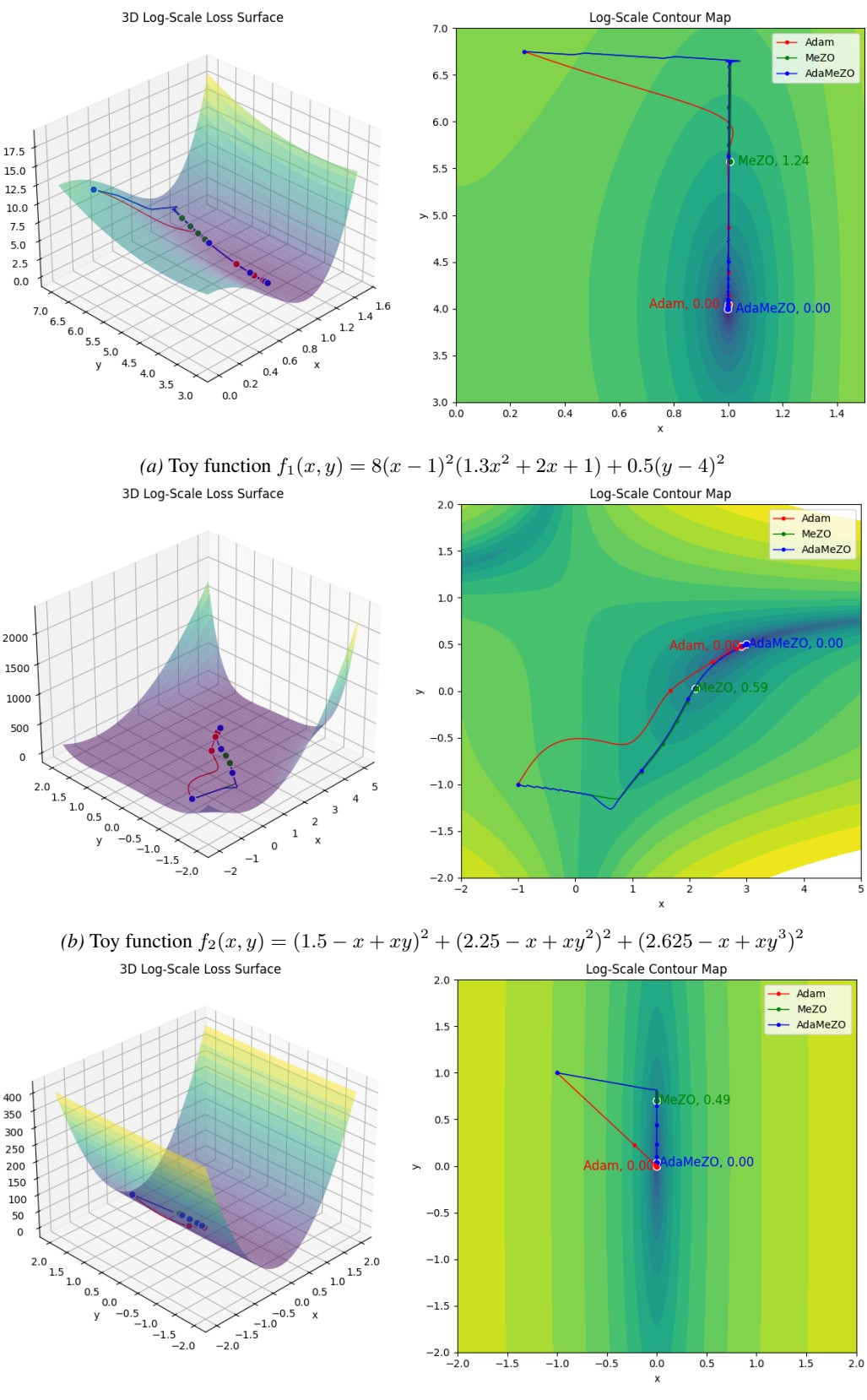

*(a)* Toy function $f_1(x, y) = 8(x-1)^2(1.3x^2 + 2x + 1) + 0.5(y-4)^2$

*(b)* Toy function $f_2(x, y) = (1.5 - x + xy)^2 + (2.25 - x + xy^2)^2 + (2.625 - x + xy^3)^2$

*(c)* Toy function $f_3(x, y) = 100x^2 + y^2$

*Figure 5.* Loss landscapes of the toy functions and optimization trajectories.

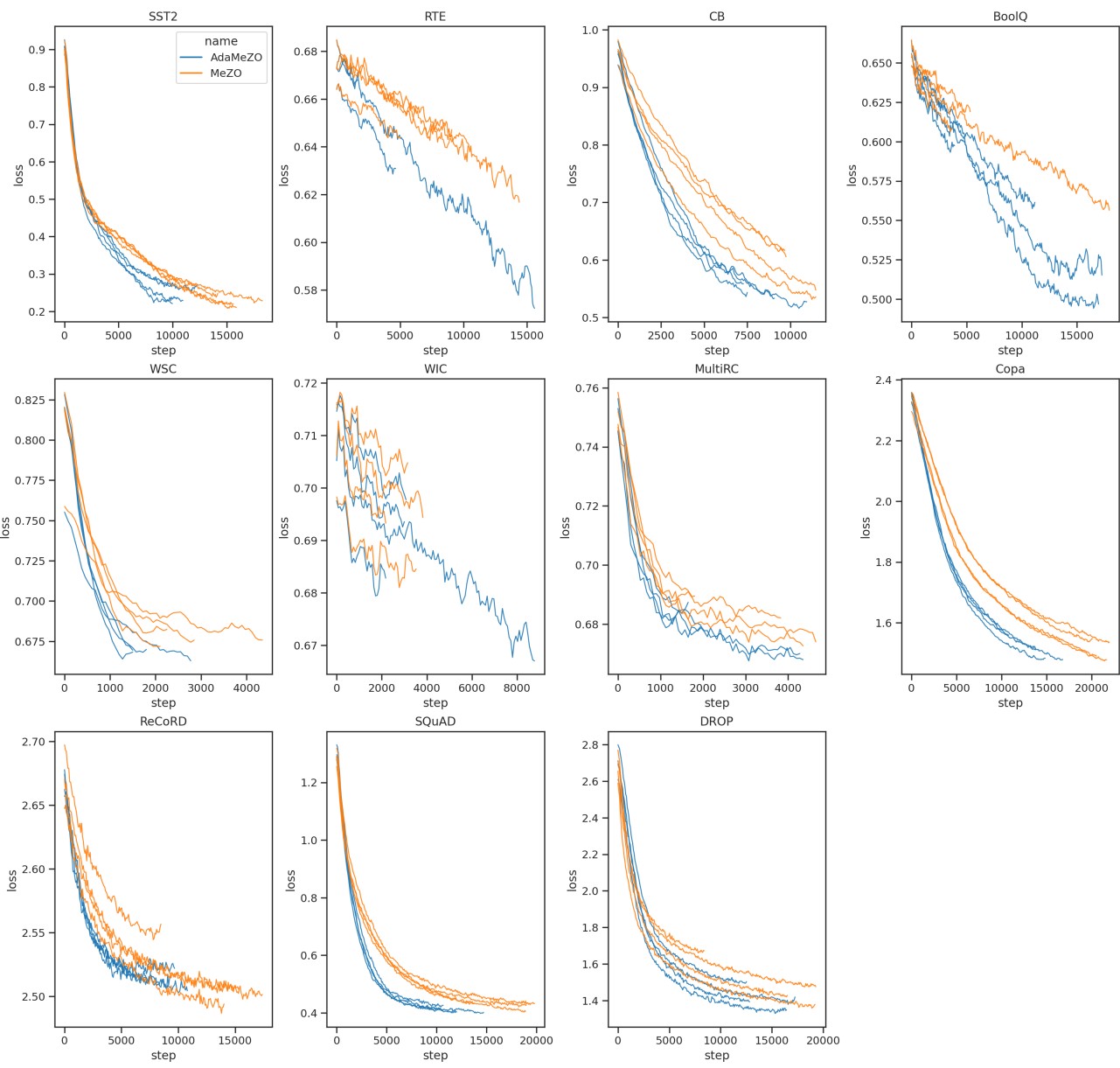

*Figure 6.* Training loss curve of OPT-13B over language tasks.

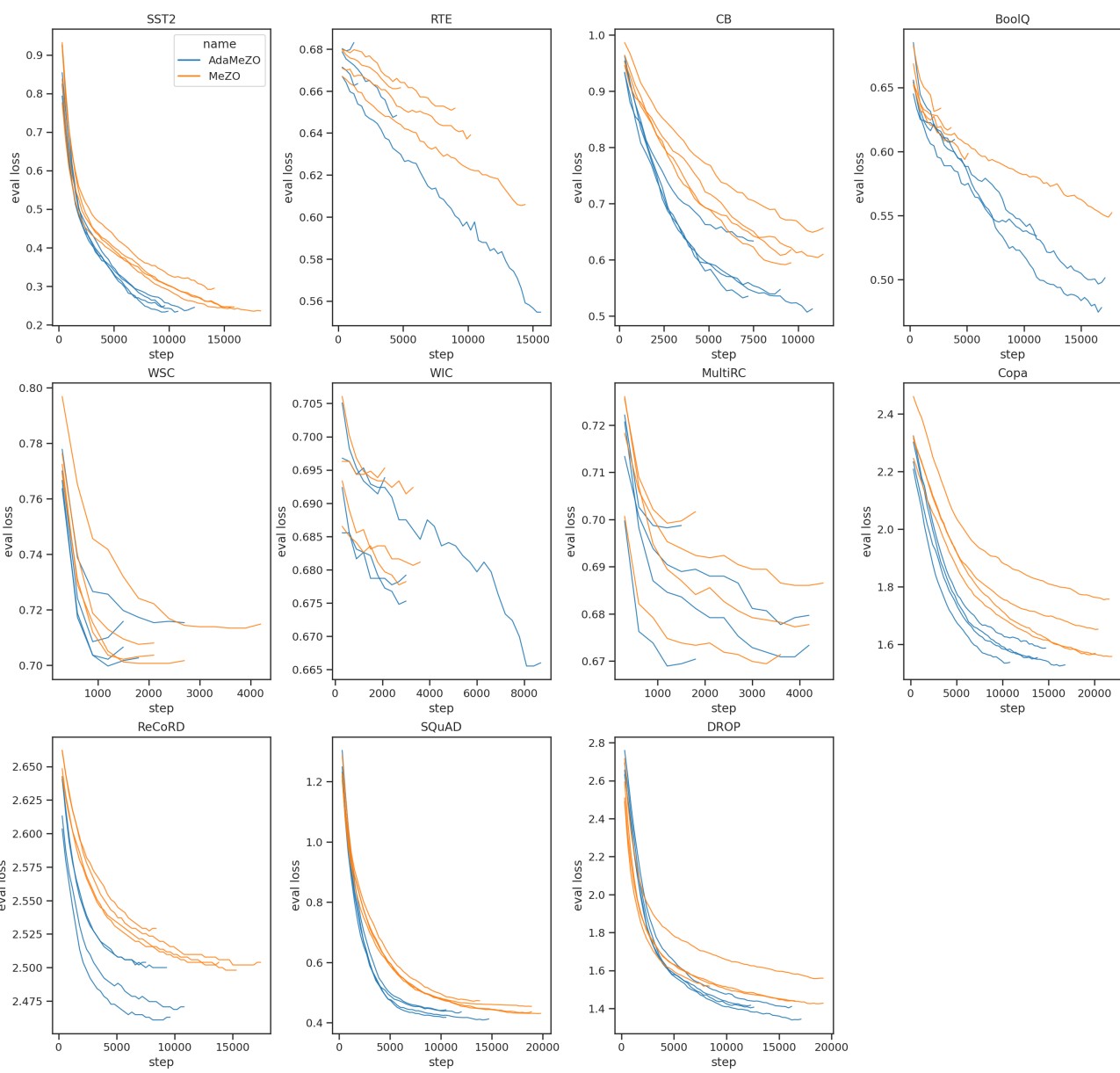

*Figure 7.* Evaluation loss curve of OPT-13B over language tasks.

*Table 14.* Hyperparameter settings for Table 6.

| Experiment | Hyperparameters | Values |
|---|---|---|
| HiZOO (prefix) | $B$ | 16 |
| | $\eta$ | $\{5e-2, 1e-2, 5e-3\}$ |
| | $\mu$ | $1e-1$ |
| | # prefix tokens | 5 |
| AdaMeZO (prefix) | $B$ | 16 |
| | $\eta$ | $\{7.5e-6, 1e-5, 2.5e-5\}$ |
| | $\mu$ | $1e-1$ |
| | # prefix tokens | 5 |

*Table 15.* Hyperparameter settings for HiZOO in Table 3, Table 5, Table 12, and Table 13.

| Experiment | Hyperparameters | Values |
|---|---|---|
| HiZOO | $B$ | 16 |
| | $\eta$ | $\{1e-6, 5e-7, 1e-7\}$ |
| | $\mu$ | $1e-3$ |

---

**Algorithm 3** AdaMeZO

---

**Input:** Initialized model parameters $\boldsymbol{w}_0 \in \mathbb{R}^d$, loss function $\mathcal{L} : \mathbb{R}^d \to \mathbb{R}$, step budget $T$, perturbation scale $\mu$, learning rate $\eta$, horizon $h$, first EMA ratio $\beta_1$, second EMA ratio $\beta_2$, block strategy $B(\boldsymbol{w}) = \{\boldsymbol{w}^{(1)}, \ldots, \boldsymbol{w}^{(b)}\}$, cancel factor $\beta_v$, warm-up steps $T_w$

**Output:** Trained model parameters $\boldsymbol{w}_T$

`seeds, projs` $\leftarrow$ `[], []`

**for** $t = 1, \ldots, T$ **do**

    Sample batch $\mathcal{B}_t$ and random seed $s$

    Reset the PRNG with random seed $s$, spawn $\boldsymbol{z}_t \sim \mathcal{N}(\boldsymbol{0}, I_d)$

    Estimate $p_t$ using Equation (1)                             # *in-place model perturbation*

    `seeds.append(`$s$`), projs.append(`$p_t$`)`

    $\boldsymbol{w}_t \leftarrow \boldsymbol{w}_t$

    **if** $t > T_w$ **then**

        `states` $\leftarrow$ `[None]` $\star$ $(h, b)$

        **for** $\tau_b = 1, \ldots, b$ **do**

            $\boldsymbol{m}, \boldsymbol{v} \leftarrow \boldsymbol{0}, \boldsymbol{0}$

            **for** $\tau_h = 1, \ldots, h$ **do**

                $p \leftarrow$ `projs[`$-\tau_h$`]`

                **if** `states[`$\tau_h, \tau_b$`]` `==` `None` **then**

                    $s \leftarrow$ `seeds[`$-\tau_h$`]`

                    Reset the PRNG with random seed $s$, spawn $\boldsymbol{z} \sim \mathcal{N}(\boldsymbol{0}, I_{|\boldsymbol{w}^{(\tau_b)}|})$

                **else**

                    Load `states[`$\tau_h, \tau_b$`]` to PRNG, spawn $\boldsymbol{z} \sim \mathcal{N}(\boldsymbol{0}, I_{|\boldsymbol{w}^{(\tau_b)}|})$

                **end if**

                Save PRNG state to `states[`$\tau_h, \tau_b$`]`

                $\boldsymbol{m} \leftarrow \boldsymbol{m} + \beta_1^{\tau_h - 1} p \boldsymbol{z}$

                $\boldsymbol{v} \leftarrow \boldsymbol{v} + \beta_2^{\tau_h - 1} p^2 (\boldsymbol{z} \odot \boldsymbol{z})$

            **end for**

        **end for**

        $\boldsymbol{w}_t^{(\tau_b)} \leftarrow \boldsymbol{w}_t^{(\tau_b)} - \eta \beta_v \frac{\boldsymbol{m}}{\sqrt{\boldsymbol{v} + \epsilon}}$

    **else**

        Reset the PRNG with random seed $s$, spawn $\boldsymbol{z} \sim \mathcal{N}(\boldsymbol{0}, I_d)$

        $\boldsymbol{w}_t \leftarrow \boldsymbol{w}_t - \eta p_t \boldsymbol{z}$

    **end if**

**end for**

---

# E. Detailed Convergence Analysis

**Lemma E.1** (Update expectations). *Given Assumption 4.2 to 4.4 and Assumption 4.5, for warm-up steps, it holds that*

$$\mathbb{E}[\boldsymbol{u}_t] = \nabla\mathcal{L}(\boldsymbol{w}_t) + \mathcal{O}(\mu), \tag{5}$$

$$\mathbb{E}[\|\boldsymbol{u}_t\|_2^2] \leq \frac{\eta^2 L\mathcal{O}(r)}{2}(\|\nabla\mathcal{L}(\boldsymbol{w}_t)\|_2^2 + \sigma^2) + \mathcal{O}(\mu^2). \tag{6}$$

*After warm-up steps, it holds that*

$$\mathbb{E}[\boldsymbol{u}_t] = \Sigma_t^{-1}(\nabla\mathcal{L}(\boldsymbol{w}_t) + \mathcal{O}(\bar{\beta}_1 L\eta)) + \mathcal{O}(\mu), \tag{7}$$

$$\mathbb{E}[\|\boldsymbol{u}_t\|_2^2] \leq (2\mathrm{tr}(\Sigma_t^{-1}) + 4s_l^{-1})(\|\nabla\mathcal{L}(\boldsymbol{w}_t)\|_{\Sigma_t^{-1}}^2 + \sigma^2 + \mathcal{O}(\bar{\beta}_1^2 L^2\eta^2)) + \mathcal{O}(\mu^2), \tag{8}$$

*where $\sigma$ captures the batch stochasticity in first-order, $\bar{\beta}_1 = 1 - \beta_1$, and $\Sigma_t$ is the diagonal matrix with $\boldsymbol{v}_t$ being its diagonal.*

*Proof.* The bounds for the warm-up phase follow Proof of Lemma 2 in (Malladi et al., 2023).

After the warm-up case, by the definition of $\boldsymbol{u}_t$, we have

$$\boldsymbol{u}_t = \Sigma_t^{-1}\boldsymbol{m}_t,$$

where

$$\Sigma_t := \beta_v \sqrt{\sum_{i=0}^{h-1} \beta_2^i \mathrm{diag}\left(\frac{1}{n}\sum_{j=1}^n \boldsymbol{g}_{t-i,j} \odot \boldsymbol{g}_{t-i,j}\right)}, \quad \boldsymbol{g}_{t,j} := \boldsymbol{z}_j^\top \nabla\mathcal{L}(\boldsymbol{w}_t, \mathcal{B}_t)\boldsymbol{z}_j,$$

with $\beta_v$ is a normalizing factor connected to $\beta_1$ and $\beta_2$ to cancel out all $\beta_1$ and $\beta_2$ related terms.

Moreover,

$$\mathbb{E}\left[\|\boldsymbol{u}_t\|_2^2\right]$$

$$=\mathbb{E}_{\mathcal{B}_t,\boldsymbol{z}_j}\left[\|\frac{1}{n}\sum_{j=1}^n \Sigma_t^{-\frac{1}{2}}\boldsymbol{z}_j\boldsymbol{z}_j^\top \Sigma^{-\frac{1}{2}}(\nabla\mathcal{L}(\boldsymbol{w}_t,\mathcal{B}_t) + \mathcal{O}(\bar{\beta}_1 L\eta)) + \mathcal{O}(\mu)\|_2^2\right]$$

$$\overset{(a)}{\leq} 2\mathbb{E}_{\mathcal{B}_t,\boldsymbol{z}}\left[\|\frac{1}{n}\sum_{j=1}^n \Sigma_t^{-\frac{1}{2}}\boldsymbol{z}_j\boldsymbol{z}_j^\top \Sigma_t^{-\frac{1}{2}}(\nabla\mathcal{L}(\boldsymbol{w}_t,\mathcal{B}_t) + \mathcal{O}(\bar{\beta}_1 L\eta))\|_2^2\right] + \mathcal{O}(\mu^2)$$

$$\overset{(b)}{\leq} \frac{2}{n}\sum_{j=1}^n \mathbb{E}_{\mathcal{B}_t,\boldsymbol{z}_j}\left[\|\Sigma_t^{-\frac{1}{2}}\boldsymbol{z}_j\boldsymbol{z}_j^\top \Sigma^{-\frac{1}{2}}(\nabla\mathcal{L}(\boldsymbol{w}_t,\mathcal{B}_t) + \mathcal{O}(\bar{\beta}_1 L\eta))\|_2^2\right] + \mathcal{O}(\mu^2)$$

$$\overset{(c)}{=} 2\mathrm{tr}(\Sigma_t^{-1})\mathbb{E}_{\mathcal{B}_j}\left[(\nabla\mathcal{L}(\boldsymbol{w}_t,\mathcal{B}_t) + \mathcal{O}(\bar{\beta}_1 L\eta))^\top \Sigma_t^{-1}(\nabla\mathcal{L}(\boldsymbol{w}_t,\mathcal{B}_t) + \mathcal{O}(\bar{\beta}_1 L\eta))\right]$$
$$+ 4\mathbb{E}_{\mathcal{B}_j}\left[(\nabla\mathcal{L}(\boldsymbol{w}_t,\mathcal{B}_t) + \mathcal{O}(\bar{\beta}_1 L\eta))^\top \Sigma_t^{-2}(\nabla\mathcal{L}(\boldsymbol{w}_t,\mathcal{B}_t) + \mathcal{O}(\bar{\beta}_1 L\eta))\right] + \mathcal{O}(\mu^2)$$

$$\overset{(d)}{\leq} (2\mathrm{tr}(\Sigma_t^{-1}) + 4s_l^{-1})\mathbb{E}_{\mathcal{B}_j}\left[(\nabla\mathcal{L}(\boldsymbol{w}_t,\mathcal{B}_t) + \mathcal{O}(\bar{\beta}_1 L\eta))^\top \Sigma_t^{-1}(\nabla\mathcal{L}(\boldsymbol{w}_t,\mathcal{B}_t) + \mathcal{O}(\bar{\beta}_1 L\eta))\right] + \mathcal{O}(\mu^2)$$

$$\overset{(e)}{\leq} (2\mathrm{tr}(\Sigma_t^{-1}) + 4s_l^{-1})(\|\nabla\mathcal{L}(\boldsymbol{w}_t)\|_{\Sigma^{-1}}^2 + \mathcal{O}(\bar{\beta}_1 L\eta) + s_l^{-1}\sigma_t^2) + \mathcal{O}(\bar{\beta}_1 L\eta).$$

where $(a)$ is by $\|a + b\|_2^2 \leq \|a\|_2^2 + \|b\|_2^2 + 2\|ab\|_2 \leq 2\|a\|_2^2 + 2\|b\|_2^2$; $(b)$ is by the convexity of the function $\|\cdot\|^2$; $(c)$ is by setting $A = \mathbb{E}_{\mathcal{B}_j}\left[\Sigma_t^{-\frac{1}{2}}(\nabla\mathcal{L}(\boldsymbol{w}_t,\mathcal{B}_t) + \mathcal{O}(\bar{\beta}_1 L\eta))^\top(\nabla\mathcal{L}(\boldsymbol{w}_t,\mathcal{B}_t) + \mathcal{O}(\bar{\beta}_1 L\eta))\Sigma_t^{-\frac{1}{2}}\right]$ and $B = \Sigma_t^{-1}$, then apply Assumption 4.5; $(d)$ is by Assumption 4.3; finally $(e)$ by Assumption 4.2. $\square$

Finally, we establish Theorem 4.7.

*Proof.* Split the full summation into the warm-up phase and the post-warm-up phase as follows.

$$\frac{1}{T}\sum_{t=1}^{T}\|\nabla\mathcal{L}(\boldsymbol{w}_t)\|_2^2 = \frac{1}{T}\underbrace{\sum_{t=1}^{T_w}\|\nabla\mathcal{L}(\boldsymbol{w}_t)\|_2^2}_{\text{warm-up}} + \frac{1}{T}\sum_{t=T_w+1}^{T}\|\nabla\mathcal{L}(\boldsymbol{w}_t)\|_2^2.$$

Choose

$$\eta \le \min\left\{\frac{1}{s(\mathrm{tr}\Sigma_t^{-1}+2s_l^{-1})\sqrt{T}}, \frac{1}{L\mathcal{O}(r)\sqrt{T}}, \frac{1}{s\mathbb{E}[\mathcal{L}(\boldsymbol{w}_1)]-\mathbb{E}[\mathcal{L}(\boldsymbol{w}_T)]\sqrt{T}}\right\},$$

Equation (5) and (6) with Assumption 4.1 yields

$$\mathbb{E}[\mathcal{L}(\boldsymbol{w}_{t+1})] \le \mathcal{L}(\boldsymbol{w}_t) - \eta\|\nabla\mathcal{L}(\boldsymbol{w}_t)\|_2^2 + \frac{\eta^2 L\mathcal{O}(r)}{2}(\|\nabla\mathcal{L}(\boldsymbol{w}_t)\|_2^2+\sigma^2) + \mathcal{O}(\mu^2)$$

$$\le \mathcal{L}(\boldsymbol{w}_t) - \frac{\eta}{2}\|\nabla\mathcal{L}(\boldsymbol{w}_t)\|_2^2 + \frac{\eta^2 L\sigma^2\mathcal{O}(r)}{2} + \mathcal{O}(\mu^2).$$

Equation (7) and (8) with Assumption 4.1 yields

$$\mathbb{E}[\mathcal{L}(\boldsymbol{w}_{t+1})] \le \mathcal{L}(\boldsymbol{w}_t) - \eta\|\nabla\mathcal{L}(\boldsymbol{w}_t)\|_{\Sigma_t^{-1}}^2 + L\eta^2(\mathrm{tr}\Sigma_t^{-1}+2s_l^{-1})\left(\|\nabla\mathcal{L}(\boldsymbol{w}_t)\|_{\Sigma_t^{-1}}^2 + \mathcal{O}(\bar{\beta}_1^2 L^2\eta^2) + \sigma^2\right) + \mathcal{O}(\mu^2)$$

$$\le \mathcal{L}(\boldsymbol{w}_t) - \frac{\eta}{2}\|\nabla\mathcal{L}(\boldsymbol{w}_t)\|_{\Sigma_t^{-1}}^2 + L\eta^2(\mathcal{O}(\bar{\beta}_1^2 L^2\eta^2)+\sigma^2)(\mathrm{tr}\Sigma_t^{-1}+2s_l^{-1}) + \mathcal{O}(\mu^2).$$

So, for the warm-up phase,

$$\frac{1}{T}\sum_{t=1}^{T_w}\|\nabla\mathcal{L}(\boldsymbol{w}_t)\|_2^2 \le \frac{2}{\eta T}(\mathcal{L}(\boldsymbol{w}_1)-\mathbb{E}[\mathcal{L}(\boldsymbol{w}_{T_w})]) + \frac{T_w L\eta\sigma^2\mathcal{O}(r)}{T} + \mathcal{O}(\mu^2), \tag{9}$$

and for the post-warm-up phase, Equation (7) and (8) with Assumption 4.1 yields

$$\frac{1}{T}\sum_{t=T_w+1}^{T}\|\nabla\mathcal{L}(\boldsymbol{w}_t)\|_2^2$$

$$\le \frac{s_u}{T}\sum_{t=T_w+1}^{T}\|\nabla\mathcal{L}(\boldsymbol{w}_t)\|_{\Sigma_t^{-1}}^2$$

$$\le \frac{2s_u}{\eta T}(\mathbb{E}[\mathcal{L}(\boldsymbol{w}_{T_w+1})]-\mathbb{E}[\mathcal{L}(\boldsymbol{w}_T)]) + \frac{s_u(T-T_w)L\eta(\mathcal{O}(\bar{\beta}_1^2 L^2\eta^2)+\sigma^2)(\mathrm{tr}\Sigma_t^{-1}+2s_l^{-1})}{T} + \mathcal{O}(\mu^2). \tag{10}$$

Take $s = \max\{1, s_u\}$, combine Equation (9) and (10),

$$e \le \frac{2s}{\eta T}(\mathbb{E}[\mathcal{L}(\boldsymbol{w}_1)]-\mathbb{E}[\mathcal{L}(\boldsymbol{w}_T)]) + \frac{T_w\eta L\sigma^2\mathcal{O}(r)}{T} + sL\eta(\mathcal{O}(\bar{\beta}_1^2 L^2\eta^2)+\sigma^2)(\mathrm{tr}\Sigma_t^{-1}+2s_l^{-1}) + \mathcal{O}(\mu^2)$$

$$\le \frac{L\sigma^2}{\sqrt{T}} + \frac{2}{T\sqrt{T}} + \frac{T_w(\sigma^2+\mathcal{O}(\bar{\beta}_1^2 L^2\eta^2))}{T\sqrt{T}} + \mathcal{O}(\mu^2).$$

We omit the higher order terms to arrive at the target. $\square$

