# OpenReview forum: "AdaMeZO: Adam-style Zeroth-Order Optimizer for LLM Fine-tuning Without Maintaining the Moments"
_ICML.cc/2026/Conference — ICML 2026 regular_

### Official Review · Reviewer_WgkV · 2026-03-06

**Soundness:** 3
**Presentation:** 4
**Significance:** 2
**Originality:** 3
**Overall Recommendation:** 5
**Confidence:** 4

**Summary:**

This paper studies memory-efficient fine-tuning of large language models using zeroth-order optimization, an Adam-style zeroth-order optimizer that avoids explicitly storing first and second moments to reduce memory overhead. The method approximates adaptive updates through truncated moment estimation and PRNG-based generation of random directions.

The paper provides theoretical convergence analysis and empirical evaluations on toy optimization problems and several LLM fine-tuning tasks. Results show that AdaMeZO improves convergence efficiency compared with MeZO while maintaining low memory usage.

**Compliance With Llm Reviewing Policy:**

Affirmed.

**Final Justification:**

I appreciate the authors’ efforts during the rebuttal process. Overall, most of my concerns have been addressed, and I will maintain my positive score.

**Key Questions For Authors:**

- The proposed PRNG mechanism seems somehow related to the idea used in FZOO[1]. It's not  necessary to include it as a baseline, but it would be helpful to discuss the differences and connections with FZOO in the related work section.

- Could you please further explain the meaning of MeZO-switch in the experiment? It's briefly mentioned in Line 365 but somehow confused.

- The paper mentions that the second-moment estimates are inexpensive but inaccurate. Is this mainly due to a trade-off between memory-efficient storage and estimation accuracy, or because the moments are computed from zeroth-order gradient estimates themselves?

[1] S. Dang, Y. Guo, Y. Zhao, H. Ye, X. Zheng, G. Dai, and I. Tsang, “Fzoo: Fast zeroth-order optimizer for fine-tuning
large language models towards adam-scale speed,”

**Limitations:**

same with the above one.

**Strengths And Weaknesses:**

Soundness: The method is technically reasonable and supported by both theoretical analysis and empirical experiments.

Presentation: The paper is generally clear and well organized, and the motivation for memory-efficient LLM fine-tuning is well explained.

Significance: Memory-efficient training and fine-tuning of LLMs is an important problem, and improving zeroth-order optimization could benefit resource-constrained settings.

Originality: The paper proposes a novel way to incorporate Adam-style adaptive updates into a zeroth-order optimizer without explicitly storing moment statistics.

---

> ### Author Rebuttal · Authors · 2026-03-31
>
> We are very happy to receive your positive review. Thanks for finding our work interesting, novel, and well presented. We will address your questions point by point in the following.
>
> **Q1 Comparison with FZOO**. Thanks for bringing this work out. FZOO and AdaMeZO share a similar philosophy: explore how cost-efficient ZO can be. For FZOO, they use one-sided Rademacher perturbations to efficiently probe a batch of random gradient directions in parallel, thereby minimizing the number of steps to convergence (possibly with additional activation memory). For AdaMeZO, we try to minimize parameter memory while maintaining the preconditioning acceleration, rather than the number of steps. We will add FZOO to the related works section in the next version.
>
> **Q2 About MeZO-switch**. Thanks for your question. We incorporate MeZO-switch to rule out the possibility that AdaMeZO is secretly MeZO with a higher learning rate. Starting from MeZO's learning rate, we gradually increase it until the trajectory of the MeZO-switch becomes longer than that of AdaMeZO. This confirms that AdaMeZO outperforms MeZO by being more clever in its exploration, rather than simply walking longer trajectories.
>
> **Q3 Vague second-moment estimation**. Thanks for your question. We believe that inaccurate second-order estimates are mainly due to zeroth-order gradient noise and to the truncated history gradient estimator used for moment estimation. It is obvious that zeroth-order gradient estimation introduces gradient noise. In addition, one can look up older gradients to get an estimate closer to what one would get with MeZO-Adam, but this affects only wall-clock time. The memory overhead is actually decoupled from the preconditioner's recovery accuracy, since it depends only on the block size.
>
> We hope that the response addresses your concerns about our work. Thank you again for your time and effort in reviewing our manuscript and providing your valuable comments.

---

> > ### Author Rebuttal · Reviewer_WgkV · 2026-04-03
> >
> > Overall, most of my concerns have been addressed, and I will maintain my positive score.

---

> > > ### Author Response · Authors · 2026-04-03
> > >
> > > We would like to thank you for your time and effort in reviewing our rebuttal. We are happy to receive your positive assessment.

---

### Official Review · Reviewer_Sru7 · 2026-03-09

**Soundness:** 3
**Presentation:** 3
**Significance:** 2
**Originality:** 2
**Overall Recommendation:** 4
**Confidence:** 3

**Summary:**

This paper introduces AdaMeZO, applying Adam-style momentum updates into MeZO with little memory increase. By leveraging truncated moment estimations and fine-scaled PRNG state caching, AdaMeZO dynamically reconstructs past gradient directions block by block, entirely bypassing the need to maintain full momentum vectors in memory.

**Compliance With Llm Reviewing Policy:**

Affirmed.

**Final Justification:**

I appreciate the authors’ efforts in the rebuttal, and all of my major concerns have been resolved. I have decided to raise my score to WA.

**Key Questions For Authors:**

1. Since a larger query budget [1] generally helps stabilize training, I would like to know whether plain MeZO could achieve comparable performance to AdaMeZO if its computational savings were used to increase the number of queries under the same training-time budget.

---
[1] Zhang Y, Li P, Hong J, et al. Revisiting zeroth-order optimization for memory-efficient llm fine-tuning: A benchmark[J]. arXiv preprint arXiv:2402.11592, 2024.

**Limitations:**

yes

**Strengths And Weaknesses:**

**Strengths**
- The main innovation of AdaMeZO lies in estimating momentum from cached random states, thereby stabilizing training while incurring negligible memory overhead.
- A detailed theoretical analysis is conducted to demonstrate the convergence of the proposed method.

**Weaknesses**
- The related work should be reorganized to place greater emphasis on ZO optimizers rather than FO optimizers.
- The experimental results seem to compare AdaMeZO only with MeZO and HiZOO, but I think it is necessary to include more ZO optimizers in the comparison, such as ZO-Adam, despite its additional memory cost.
- The time-step notation $t$ is confused in most equations. For example, in $m_t \leftarrow \beta_1 m_t + (1-\beta_1) g_t$, the $m_t$ on the right-hand side should be $m_{t-1}$.
- In Equation 2, the superscript on $\beta_1$ is incorrect.
- The optimization curves in most figures should include more baselines to better demonstrate the superiority of the proposed method.

---

> ### Author Rebuttal · Authors · 2026-03-31
>
> Thanks for your time and effort in reviewing our work and finding it innovative and detailed. We will address your concerns point by point in the following.
>
> **W1 Need more ZO works in the related works section**. Thanks for your comment. We will move the detailed related works from the Appendix to the main text's related works section.
>
> **W2 More baselines**. Thanks for your comment. We will bring MeZO-Adam into comparison. Some preview results are as follows. We will provide a full comparison in the next version.
>
> |Optimizer\Task| SST2 | RTE | CB | BoolQ | WSC | WIC | MultiRC | Copa | ReCoRD | SQuAD | DROP |
> |---:|---:|---:|---:|---:|---:|---:|---:|---:|---:|---:|---:|
> |MeZO|90.9 (0.3)|52.5 (1.5)|65.5 (6.9)|61.8 (2.1)|51.1 (8.4)|58.6 (1.4)|53.7 (2.2)|74.5 (3.6)|70.6 (1.0)|73.3 (0.2)|22.8 (0.6)|
> |HiZOO|90.9 (1.0)|**54.5** (1.6)|63.3 (8.5)|***62.7*** (1.6)|49.4 (6.9)|***58.4*** (0.4)|***55.4*** (1.7)|74.0 (1.8)|70.8 (0.8)|74.5 (0.4)|24.5 (0.5)|
> |MeZO-Adam|**91.8** (0.3)|53.6 (0.3)|**70.5** (1.0)|61.5 (0.5)|***52.1*** (3.5)|57.7 (0.3)|54.9 (1.0)|**76.3** (1.3)|**71.5** (0.4)|***75.4*** (0.8)|**24.7** (0.7)|
> |AdaMeZO|***91.6*** (0.3)|***54.3*** (3.1)|***69.6*** (1.4)|**63.2** (1.6)|**53.5** (7.8)|**58.4** (1.6)|**55.9** (0.7)|***75.5*** (4.0)|***71.1*** (1.3)|**76.1** (0.7)|***24.6*** (1.0)|
>
> For MeZO-Adam, we tested $lr=\{1e-5, 1e-6, 1e-7\}$, $B=16$, $\mu=1e-3$, $\beta=\{(0.7, 0.9), (0.9, 0.999)\}$. **Bold** and ***Italic bold*** denotes **best** and ***second-best***, respectively. We also provide a runtime \& memory profile to demonstrate the cost-effectiveness of AdaMeZO.
>
> |Optimizer\Runtime (sec/step)| SST2 | Copa | SQuAD |
> |---:|---:|---:|---:|
> |MeZO|0.21|0.18|0.21|
> |HiZOO|0.23|0.23|0.24|
> |MeZO-Adam|0.22|0.22|0.20|
> |Adam|0.12|0.13|0.13|
> |AdaMeZO|0.31|0.30|0.31|
>
> |Optimizer\Memory (MB)| SST2 | Copa | SQuAD | Ratio |
> |---:|---:|---:|---:|---:|
> |MeZO|5016|5058|5040|1x|
> |HiZOO|7532|7535|7396|1.49x|
> |MeZO-Adam|9481|9465|9555|1.89x|
> |Adam|22172|21660|22688|4.40x|
> |AdaMeZO|5410|5452|5434|1.07x|
>
> **W3 Unclear subscripts**. Thanks for your comment. We will revise the subscripts throughout the paper to improve readability.
>
> **W4 Incorrect superscript**. Thanks for pointing this out. It should be $\beta_1^{h-1}$ rather than $\beta_1^{t-h-1}$. We will correct this in our next version.
>
> **W5 Need more baselines**. Thanks for your comment. We will add more baselines to the loss curves in the next version.
>
> **Q1 Multi-query MeZO comparison**. Thanks for your question. We report a comparison as follows.
>
> |Optimizer\Task acc ↑ | SST2 | Copa | SQuAD |
> |---:|---:|---:|---:|
> |MeZO|***90.9*** (0.3)|***74.5*** (3.6)|***73.3*** (0.2)|
> |MeZO(#query=2)|88.8 (1.8)|74.5 (1.3)|71.2 (0.4)|
> |AdaMeZO|**91.6** (0.3)|**75.5** (4.0)|**76.1** (0.7)|
>
> |Optimizer\Runtime ↓ (sec/step) | SST2 | Copa | SQuAD |
> |---:|---:|---:|---:|
> |MeZO|**0.21**|**0.18**|**0.21**|
> |MeZO (#query=2)|0.34|0.33|0.37|
> |AdaMeZO|***0.31***|***0.30***|***0.31***|
>
> This echoes Malladi et al.'s conclusion from the ablation experiment on sample schedules in their Section A.2 in [ref1]: an increasing number of queries yields only marginal gains at best.
>
> **Reference**
>
> [ref1] Malladi et al., Fine-Tuning Language Models with Just Forward Passes.
>
> ***
>
> We hope that the response addresses your concerns about our work. Thank you again for your time and effort in reviewing our manuscript and providing your valuable comments.

---

> > ### Author Rebuttal · Reviewer_Sru7 · 2026-04-01
> >
> > I appreciate the authors’ efforts in the rebuttal, and all of my major concerns have been resolved. Although some minor revisions could not be presented immediately, I am still willing to raise my score to a positive rating.

---

> > > ### Author Response · Authors · 2026-04-03
> > >
> > > We would like to thank you for your time and effort in reviewing our rebuttal. We are happy to receive your positive assessment.

---

### Official Review · Reviewer_rria · 2026-03-10

**Soundness:** 3
**Presentation:** 3
**Significance:** 2
**Originality:** 2
**Overall Recommendation:** 3
**Confidence:** 4

**Summary:**

The paper proposes a memory-efficient Adam-style zero-order optimizer that computes the conditioner block-wise via clever random-state caching. The method demonstrates improvements over existing approaches across extensive experiments and provides convergence guarantees in the classical L-smooth setting.

**Compliance With Llm Reviewing Policy:**

Affirmed.

**Final Justification:**

I thank the authors for their rebuttals. However, I still find that the work offers limited theoretical and practical novelty, and thus its overall significance remains modest. Therefore, I would like to maintain my current score.

**Key Questions For Authors:**

1. Could the authors clarify what exactly is being analyzed in the theoretical section?
2. Is there any particular reason for not including a comparison with Helene?
3. What does $\mu^2$ represent in the convergence bound? How does the bound depend on $\sigma$?

**Limitations:**

Yes.

**Strengths And Weaknesses:**

### Strengths

1. The paper provides extensive experiments on language fine-tuning tasks and reports improvements over existing methods.

### Weaknesses

1. A clearer presentation of how momentum is recovered through state caching would make the method easier for readers to understand.
2. It is somewhat surprising that the method outperforms HiZoo, which is a Hessian-informed zero-order method that uses more memory.
3. The idea of computing the preconditioner block-wise appears fairly standard and not particularly original, as similar approaches have been used in many first-order optimization algorithms.
4. The theoretical section would benefit from additional clarification. For example, is the algorithm equivalent to zero-order Adam in this setting, or should it be viewed as an approximation of it?

### Minor Comments

1. *“Adam’s great performance could be attributed to its similarity to sign descent with momentum.”*
   I do not believe this is the claim made by Kunstner et al. Rather, they argue that sign gradient descent is a useful proxy for studying Adam, not that Adam performs well because it resembles sign gradient descent.

---

> ### Author Rebuttal · Authors · 2026-03-31
>
> Thanks for your time and effort reviewing our work. We will address your concerns point by point in the following.
>
> **W1 Clearer presentation**. Thanks for your comment. The key idea behind using a block-wise preconditioner is the form of the Adam update: $m / \sqrt{v + \epsilon}$, which is a non-linear combination of history gradients $g_t$'s. In our work, they are treated like $(g_t + ... + \beta_1^h g_{t-h}) / \sqrt{g_t^2 + ... + \beta_2^h g_{t-h}^2 + \epsilon}$. Since it is unable to compute this term in-place due to non-linearity, the classic approach would be materialize and store $v = g_t^2 + ... + \beta_2^h g_{t-h}^2 + \epsilon$, then reconstruct $g_t / \sqrt{v + \epsilon}$, ..., $\beta_1^h g_{t-h} / \sqrt{v + \epsilon}$ and modify the model parameters in-place.
>
> Our key idea is to avoid explicitly storing these buffers by reconstructing them block by block via the deterministic recoverability of modern PRNGs from random states. This enables memory-efficient preconditioner recovery with marginal extra memory.
>
> We will revise this part accordingly.
>
> **W2 Outperforming HiZOO**. Thanks for your comment. We would like to point out that, unlike AdaMeZO, HiZOO considers only the second-order moment, which may partially account for the performance gap. Additionally, AdaMeZO simply finds a more efficient way to represent and use first- and second-order information, rather than discarding them. Nevertheless, HiZOO still remains more competitive for multiple-choice tasks, as in Table 5.
>
> **W3 Block-wise preconditioner**. Thank you for the comment. We agree that block-wise preconditioning itself is not new. Our novelty is not the block structure itself, but rather using it to reduce memory overhead during preconditioner reconstruction by harnessing PRNGs. This differs from prior memory-saving approaches such as quantization or compression.
>
> **W4 Positioning of AdaMeZO**. Thank you for the comment. AdaMeZO is an approximation to MeZO-Adam, i.e., Adam-style optimization built on ZO gradients. We will clarify this in the next version.
>
> **W5 On Kunstner et al.**. Thank you for the correction. We agree that our original wording was too strong. Kunstner et al. did not claim that Adam performs well because it resembles sign descent with momentum. Rather, their point is that sign-based methods provide a useful proxy or analytical lens for understanding some aspects of Adam's behavior. We will revise it accordingly to avoid suggesting a causal attribution.
>
> **Q1 Analysis objective clarification**. Thank you for the question. We analyze the method's convergence to a first-order stationary point ($\|\nabla L(w)\|_2^2 = 0$) in a standard nonconvex sense. Rather than certifying global optimality, the result shows that the algorithm drives the model iterates toward a small region near a specific local optimum: R.H.S. $\to const.$ as $T \to \infty$, guarantees that $\|\nabla L(w_T)\|_2^2 \to 0$ as $T \to \infty$. Otherwise, we will have $\|\nabla L(w_t)\|_2^2 \geq const.$ for all $t > T$ with any $T$ implying divergence. We will formally introduce the target in the next version.
>
> **Q2 Helene comparison**. Thanks for your question. The authors of HELENE did not release their code, so we did not include it in the comparison. In light of your comment, we implemented MeZO-Adam as a baseline, which we believe shares the same spirit as HELENE. Some preview results are as follows. We will provide full results in the next version.
>
> |Optimizer\Task| SST2 | RTE | CB | BoolQ | WSC | WIC | MultiRC | Copa | ReCoRD | SQuAD | DROP |
> |---:|---:|---:|---:|---:|---:|---:|---:|---:|---:|---:|---:|
> |MeZO|90.9 (0.3)|52.5 (1.5)|65.5 (6.9)|61.8 (2.1)|51.1 (8.4)|58.6 (1.4)|53.7 (2.2)|74.5 (3.6)|70.6 (1.0)|73.3 (0.2)|22.8 (0.6)|
> |HiZOO|90.9 (1.0)|**54.5** (1.6)|63.3 (8.5)|***62.7*** (1.6)|49.4 (6.9)|***58.4*** (0.4)|***55.4*** (1.7)|74.0 (1.8)|70.8 (0.8)|74.5 (0.4)|24.5 (0.5)|
> |MeZO-Adam|**91.8** (0.3)|53.6 (0.3)|**70.5** (1.0)|61.5 (0.5)|***52.1*** (3.5)|57.7 (0.3)|54.9 (1.0)|**76.3** (1.3)|**71.5** (0.4)|***75.4*** (0.8)|**24.7** (0.7)|
> |AdaMeZO|***91.6*** (0.3)|***54.3*** (3.1)|***69.6*** (1.4)|**63.2** (1.6)|**53.5** (7.8)|**58.4** (1.6)|**55.9** (0.7)|***75.5*** (4.0)|***71.1*** (1.3)|**76.1** (0.7)|***24.6*** (1.0)|
>
> For MeZO-Adam, we tested $lr=\{1e-5, 1e-6, 1e-7\}$, $B=16$, $\mu=1e-3$, $\beta=\{(0.7, 0.9), (0.9, 0.999)\}$. **Bold** and ***Italic bold*** denotes **best** and ***second-best***, respectively.
>
> **Q3 Parameter definition**. Thanks for your question. $\mu$ is the perturbation scale, as defined in Eq (1). The $\sigma$ is originally in $O(1/T)$ term, so it is suppressed by the $O(1/\sqrt{T})$ term and does not appear in the theorem in the initial version. In light of your comment, we will provide a more detailed upper bound in the next version.
>
> ***
>
> We hope that the response addresses your concerns about our work. Thank you again for your time and effort in reviewing our manuscript and providing your valuable comments.

---

> > ### Author Rebuttal · Reviewer_rria · 2026-04-01
> >
> > Thanks for addressing the concerns. I appreciate the effort, but I believe the contribution is primarily engineering-focused and lacks substantial novelty. The empirical improvements over the baselines are marginal, and there is no meaningful theoretical advancement. Therefore, I would like to maintain my original score.

---

> > > ### Author Response · Authors · 2026-04-03
> > >
> > > We thank the reviewer for the detailed assessment and for clarifying the remaining concerns. We understand that the reviewer continues to view the work as primarily engineering-focused, with modest empirical gains and limited theoretical novelty. We would like to respond to the concerns point by point.
> > >
> > > **The characterization of the contribution is primarily engineering-focused.**
> > > The core contribution is not a purely implementation-level optimization, but a different algorithmic design for memory-efficient zeroth-order optimization based on in-place, on-the-fly moment reconstruction. This design enables Adam-style updates under a substantially tighter memory budget, as also supported by the reported memory and runtime profiles. In our view, the key challenge is algorithmic: how to preserve the desired update behavior without explicitly materializing the heavy optimizer states, which represents a methodology shift.
> > >
> > > **Novelty.**
> > > Our claim is not that the paper introduces a fundamentally new optimization paradigm. Rather, the novelty lies in a novel and practically relevant mechanism for reconciling Adam-style optimization with strict memory constraints in the zeroth-order setting. We believe this design point is nontrivial and distinct from conventional memory-saving approaches, such as quantization or compression (after materializing a true value), and, to the best of our knowledge, has not been addressed in this form by prior work. The paper also provides a standard theoretical characterization to support this algorithmic design.
> > >
> > > **Empirical improvements are marginal.**
> > > We agree that the absolute performance margins are not dramatic in every setting. However, we would like to emphasize that the appropriate evaluation criterion in this work is the performance-memory trade-off, rather than absolute accuracy alone. In the targeted memory-constrained zeroth-order fine-tuning regime, achieving on-par or marginally higher performance compared to strong baselines while using substantially less memory is, in our view, a meaningful improvement. A more comprehensive interpretation of the numerical gaps is that they are obtained under a substantially tighter memory budget, as shown by memory and runtime profiles.
> > >
> > > **Theoretical advancement.**
> > > We agree that the paper does not aim to introduce a fundamentally new theoretical framework. The role of the theory is to place the proposed method on a standard analytical footing by establishing a classical nonconvex convergence guarantee, rather than to claim a major advance in optimization theory. The paper's main contribution is algorithmic, while the theoretical analysis provides supporting evidence for the method's soundness.
> > >
> > > ***
> > >
> > > We respect the reviewer’s final assessment, but hope the above clarification helps more precisely position our work's novelty and contribution. We thank the reviewer again for the time and careful consideration.

---

### Official Review · Reviewer_ypfs · 2026-03-12

**Soundness:** 3
**Presentation:** 3
**Significance:** 2
**Originality:** 2
**Overall Recommendation:** 4
**Confidence:** 4

**Summary:**

This paper studies memory-efficient zeroth-order optimization for LLM fine-tuning. It proposes AdaMeZO, an extension of MeZO that introduces Adam-style first- and second-moment estimation without explicitly storing momentum states. The key idea is to reconstruct truncated exponential moving averages through PRNG state caching, aiming to maintain the low memory footprint of MeZO while improving optimization behavior. The authors provide a convergence analysis and evaluate the method on several LLMs, varying from 1.3B to 30B parameters.

**Compliance With Llm Reviewing Policy:**

Affirmed.

**Final Justification:**

I appreciate the authors' efforts in the rebuttal. My major concerns are addressed. I raise my score to encourage the authors to polish the final version carefully.

**Key Questions For Authors:**

- The authors are encouraged to improve the quality of the figures and the formatting of the appendix. When computational resources allow, confidence intervals should be reported for the experimental results, similar to the presentation in Figure 6.

- In Figure 6, the variance of the curves appears to increase noticeably toward the end of training. It would be helpful if the authors could clarify whether this is due to insufficient training length. Extending the training period may help better understand this behavior.

- The proposed method seems to be trading space for time. If the improvement strategy described in Section 5.4 is applied, would it introduce additional memory overhead and potentially contradict the original motivation of reducing memory usage?

**Limitations:**

Yes.

**Strengths And Weaknesses:**

Strengths:

- The paper studies an important problem, namely, improving the efficiency of zeroth-order optimization for LLM fine-tuning under limited memory constraints.

- The paper is generally well presented, and the intuition behind the proposed method is relatively clear and easy to follow.

- The empirical results indicate improvements over MeZO and related variants across several benchmarks while maintaining low memory usage.

Weaknesses:

- The proposed truncated moment reconstruction and gradient recovery provide an interesting way to introduce adaptive moment estimation without explicitly storing full optimizer states. However, this does not seem to introduce a fundamentally new optimization principle or design insight, and the contribution is largely an engineering implementation. As a result, the overall significance of the method appears somewhat limited.

- The theoretical analysis follows standard convergence arguments and does not appear to provide substantially new theoretical insights. Given this, the authors may consider reducing the length of the theoretical section. Instead, it may be more beneficial to improve the clarity of the presentation of the block-wise and reconstruction techniques that the method relies on, and reconsider the presentation of Figure 2 for better readability.

- The introduction of SPSA perturbations in Section 3.1 should be treated more carefully. Simply decreasing the perturbation magnitude reduces the bias of the gradient estimator but can significantly increase its variance, which may negatively affect the overall algorithm. Convergence is typically ensured through an appropriate coupling between the perturbation scale and the step-size schedule. The authors may refer to classical results in the stochastic optimization/approximation literature.

---

> ### Author Rebuttal · Authors · 2026-03-31
>
> Thanks for finding our work interesting, well-presented, and practical. We will address your concerns point by point in the following.
>
> **W1 Novelty of moment reconstruction**. Thanks for your comment. We agree that block-wise is not new in itself. However, in contrast to traditional memory-saving methods, such as compressing or quantizing the moments, we explored the new idea of approximately reconstructing the moments in-place by harnessing modern PRNGs, so that moments can be maintained with marginal additional memory.
>
> With this design, AdaMeZO achieved competitive performance using inference-level memory, meanwhile, avoiding significant increases in wall-clock time. Moreover, the design can be extended to other Adam-like zeroth-order optimizers that use preconditioners for better convergence.
>
> **W2 Clearer explanation of the core design**. Thanks for your comment. We agree that the theoretical section follows standard non-convex analysis rather than a new general convergence technique. The claim we meant to make in the section is that AdaMeZO converges as well as optimizers that use substantially more memory, shown by classic convergence analysis. We will enhance the design section and improve the readability of Figure 2 in light of your comment.
>
> **W3 More careful preliminary introduction**. Thanks for your insightful comment. We agree that simply shrinking $\mu$ is not always beneficial. This can be shown by noticing that $g_t = \frac{L(w_{t-1} + \mu z_t, B_t) - L(w_{t-1} - \mu z, B_t)}{2 \mu} z_t=  \frac{L(w_{t-1} + \mu z_t) - L(w_{t-1} - \mu z)}{2 \mu} z_t + \frac{\xi}{2 \mu} z_t$, where $\xi$ is the batch sampling noise term, since batch gradients is only estimates of true gradients, following [ref1].
>
> On the right-hand side of the equation, we have the first term as the desired signal term, and the second term depicts the variation caused by batch sampling and perturbation scale $\mu$, with $E[||{\xi z_t}/{2 \mu} ||_2^2] = O(\sigma^2 / B \mu^2)$. So there will be an additional term $O(\sigma^2 / B \mu^2)$ in Eq. (6), and finally the overall bound will be in the form of $O(1 / \sqrt{T}) + O(\mu^2) + O(\sigma^2 / B \mu^2)$. This will reflect the negative effect of having an overly small $\mu$.
>
> We maintain a constant learning rate, consistent with prior works and empirical results. The fix is minimal by replacing $O(\mu^2)$ with $O(\mu^2) + O(\sigma^2 / B \mu^2)$ and does not affect the claim of $O(1 / \sqrt{T})$ convergence.
>
> We are grateful for this valuable comment.
>
> **Q1 Add confidence intervals in figures**. Thanks for the comment. We will optimize the figures, using Figure 6 as an example, to the best of our ability.
>
> **Q2 Increasing variance in the late stage of training**. Thanks for the comment. We will provide training curves with more steps in the next version.
>
> **Q3 Potential improvement**. Thanks for the question. We believe that it would not contradict the memory-efficient advantage. These improvement strategies are based on Table 7, which shows that AdaMeZO spends most of its time performing gradient accumulation rather than additional PRNG calls. Hence,
> - By "optimizing this accumulation process", we refer to using faster addition operators or better computational devices for faster gradient accumulation during model parameter modification.
> - By "using prefix tuning", we refer to finetuning parts of the models, rather than in full-parameter. This would speed up AdaMeZO because the history gradient accumulation occurs only on part of the model, reducing the intensive addition and potentially resulting in lower final performance. We did not expand this part and kept our investigation under the full-parameter setting in line with prior work.
>
> We will revise this part in the next version.
>
> **Reference**
>
> [ref1] Spall J C. Implementation of the simultaneous perturbation algorithm for stochastic optimization[J]. IEEE Transactions on aerospace and electronic systems, 1998, 34(3): 817-823.
> ***
>
> We hope that the response addresses your concerns about our work. Thank you again for your time and effort in reviewing our manuscript and providing your valuable comments.

---

> > ### Author Rebuttal · Reviewer_ypfs · 2026-04-04
> >
> > I appreciate the authors' efforts in the rebuttal. My major concerns are addressed. I will raise my score to a positive rating and expect the authors to properly implement the corresponding revisions in the final version.

---

> > > ### Author Response · Authors · 2026-04-04
> > >
> > > Thank you for your time and effort in reviewing our rebuttal. We are happy to receive your positive assessment. We will implement the corresponding part in the final version.

---

### Decision · Program_Chairs · 2026-04-30

**Decision:**

Accept (regular)

**Comment:**

This paper introduces AdaMeZO, a memory-efficient zeroth-order optimizer for LLM fine-tuning that combines truncated moment estimation with PRNG state caching to recover Adam-style first- and second-moment information without explicitly storing full optimizer states. The problem is important, the core design is practically meaningful, and the empirical results are encouraging: across RoBERTa, OPT, and LLaMA settings, the method generally improves over MeZO-family baselines while keeping memory usage close to MeZO and far below standard Adam-style alternatives.

At the same time, the paper is best viewed as an algorithmic contribution rather than a major conceptual or theoretical advance. The main remaining concern in the reviews was about novelty/significance, with one reviewer continuing to view the work as too engineering-focused; however, the rebuttal clarified the intended contribution, strengthened the empirical positioning, and resolved the major concerns of the more positive reviewers. Taking the reviews and discussion together, I am supportive overall. For the final version, I strongly encourage the authors to improve the exposition of the block-wise/state-caching mechanism, more clearly position the theory as a standard nonconvex convergence characterization supporting the method, and strengthen the discussion of related zeroth-order baselines and comparisons.